# Aerosol indirect effects on the nighttime Arctic Ocean surface from thin, predominantly liquid clouds

Lauren M. Zamora[1,2*], Ralph A. Kahn[2], Sabine Eckhardt[3], Allison McComiskey[4], Patricia Sawamura[5,6], Richard Moore[5], Andreas Stohl[3]

[1]former NASA Postdoctoral Program Fellow, Universities Space Research Association; Now at Earth System Science Interdisciplinary Center (ESSIC), University of Maryland, College Park, MD, USA

[2]NASA Goddard Space Flight Center, Greenbelt, MD, USA

[3]NILU - Norwegian Institute for Air Research, Kjeller, Norway

[4]NOAA Earth System Research Laboratory, Boulder, CO, USA

[5]NASA Langley Research Center, Hampton, VA, USA

[6]Science Systems and Applications, Inc., Greenbelt, MD, USA

*Correspondence to*: Lauren M. Zamora (lauren.m.zamora@nasa.gov)

**Abstract.** Aerosol indirect effects have potentially large impacts on the Arctic Ocean surface energy budget, but model estimates of regional-scale aerosol indirect effects are highly uncertain and poorly validated by observations. Here we demonstrate a new way to quantitatively estimate aerosol indirect effects on a regional scale from remote sensing observations. In this study, we focus on nighttime, optically thin, predominantly liquid clouds. The method is based on differences in cloud physical and microphysical characteristics in carefully selected clean, average and aerosol-impacted conditions. The cloud subset of focus covers just ~5% of cloudy Arctic Ocean regions, warming the Arctic Ocean surface by ~1-1.4 W m$^{-2}$ regionally during polar night. However, within this cloud subset, aerosol and cloud conditions can be determined with high confidence using CALIPSO and CloudSat data and model output. This cloud subset is generally susceptible to aerosols, with a polar nighttime estimated maximum regionally integrated indirect cooling effect of ~ -0.11 W m$^{-2}$ at the Arctic sea ice surface (~10% of the clean background cloud effect), excluding cloud fraction changes. Aerosol presence is related to reduced precipitation, cloud thickness, and radar reflectivity, and in some cases, an increased likelihood of cloud presence in the liquid phase. These observations are inconsistent with a glaciation indirect effect and are consistent with either a deactivation effect or less efficient secondary ice formation related to smaller liquid cloud droplets. However, this cloud subset shows large differences in surface and meteorological forcing in shallow and higher altitude clouds and between sea ice and open ocean regions. For example, optically thin, predominantly liquid clouds are much more likely to overlay another cloud over the open ocean, which may reduce aerosol indirect effects on the surface. Also, shallow clouds over open ocean do not appear to respond to aerosols as strongly as over stratified sea ice environments, indicating a

larger influence of meteorological forcing over aerosol microphysics in these types of clouds over the rapidly changing Arctic Ocean.

## 1 Introduction

Aerosol indirect effects on clouds are among the biggest uncertainties in climate models (Boucher et al., 2013). It is particularly important to reduce these uncertainties in the Arctic, where warming is occurring at a faster rate than in other locations (Serreze et al., 2009), and where local aerosol indirect effects can be large (Garrett et al., 2004; Garrett and Zhao, 2006; Lubin and Vogelmann, 2006; Zhao and Garrett, 2015). Understanding aerosol indirect effects is also important because aerosol emissions within and in the vicinity of the Arctic are changing, and perhaps more importantly, the major aerosol removal processes and transport pathways to the Arctic may be changing as well (Jiao and Flanner, 2016).

Unfortunately, accurate observation-based estimates of regional mean forcings are very difficult to obtain at most locations around the planet due to a variety of confounding factors and errors. These include: 1) a reliance on proxies for cloud condensation nuclei (CCN) and ice nucleating particles (INP), 2) meteorological co-variability and other synoptic-scale surface and atmospheric factors, including the aerosol spatial distribution, 3) the complexity of cloud responses to aerosol type and amount (Fan et al., 2016), 4) spatial and temporal limitations of the datasets, and 5) an insufficient understanding of cloud characteristics even in the absence of anthropogenic aerosols (Ghan et al., 2016; Wilcox et al., 2015). Knowledge of this last factor is difficult to obtain because pristine conditions are rare in most locations globally (Hamilton et al., 2014). To quantify mean regional aerosol indirect effects using observations, one would need datasets that cover the large spatial and temporal scales required to include the full range of natural heterogeneity, plus a way to correctly identify clean background conditions. As a result, current estimates of regional indirect aerosol impacts on the surface radiation rely predominantly upon models that still cannot accurately represent many relevant Arctic processes (e.g., Morrison et al. (2012); Ovchinnikov et al. (2014)).

In some ways, isolating aerosol indirect effects over the Arctic Ocean can be even more challenging than in other regions. Sampling conditions at the ground are harsh, there is low thermal and visible contrast between sea ice and clouds, and observations are limited by the frequent presence of multi-layer clouds. The very cold temperatures that characterize the Arctic affect chemical reactions and physical processes (e.g., the development of frost flowers, diamond dust, and blowing snow), making comparisons with lower latitude systems more challenging. However, the Arctic Ocean is ideal for the study of indirect effects in other ways. For example, the surface and meteorological conditions over sea ice are highly homogenous compared to many other regions of the world. Moreover, pristine conditions still occur in this region with relatively high frequency, despite periodic episodes of combustion-derived aerosol transport from lower latitudes. Current day observations in clean background conditions are among our best proxies for pre-industrial conditions (Hamilton et al., 2014), and a better

understanding of pre-industrial conditions is, in turn, key to the ability to determine current-day indirect aerosol impacts on a regional scale (e.g., Gettleman (2015); Ghan et al. (2016; 2013); Carslaw et al. (2013); Wilcox et al. (2015); Kiehl et al. (2000)).

Here we present a method for identifying spatially distributed properties in a subtype of Arctic Ocean clean background clouds using a combination of the CALIPSO and CloudSat active remote sensing instruments and an atmospheric transport model. We use the difference between average cloud characteristics gathered across the Arctic Ocean and average clean background clouds over the same region to estimate the maximum regional indirect aerosol impacts on the surface. This calculation provides an estimate of the actual regional impact of aerosol indirect effects on the surface including aerosol-

meteorological co-variability after stochastic meteorological effects have been taken into account. We also examine differences between the cloud characteristics under various aerosol conditions to assess cloud formation mechanisms in the presence of aerosol.

    One goal of this work is to illustrate one way that regional-scale aerosol indirect effects on the surface can be obtained

quantitatively from observational data. In the past, such estimates have primarily been supplied only by models. We focus on the subset of Arctic Ocean clouds where aerosol impacts can be identified with the greatest certainty: optically thin (cloud optical depth, COD < 3), predominantly liquid clouds during polar night.  Optically thin, liquid-containing clouds are generally common over this region (Bennartz et al., 2013; Shupe and Intrieri, 2004). Such clouds are also effective at radiating longwave (LW) radiation downward (e.g., Garrett and Zhao (2006)), thus having a potentially large contribution to

surface forcing (Shupe and Intrieri, 2004). Moreover, models tend to under-predict the formation of these optically thin clouds at supercooled temperatures (Cesana et al., 2012), making aerosol influences on droplet characteristics and ice nucleation of particular interest.  Within the larger liquid-containing cloud group, this study focuses on predominantly liquid clouds, where aerosol conditions can be assessed with highest certainty. The analysis is also limited to nighttime samples both to improve CALIPSO aerosol-condition assessments and to reduce confounding impacts from direct and semi-direct

effects.

## 2 Methods

### 2.1 Sample selection

To describe aerosol impacts on Arctic Ocean clouds with high confidence using CALIPSO and CloudSat data, it was vital that we be able to identify clean background cases accurately. We selected a specific group of clouds where non-background

aerosol (hereafter simply referred to as "aerosol") conditions and cloud properties could be ascertained with the greatest confidence. The main Arctic Ocean cloud subset of focus consists of clouds that are Optically thin (COD < ~3), Nighttime, predominantly Liquid clouds, henceforth referred to as "ONLi" clouds for brevity. Because the ONLi cloud profiles were

taken only at night, the majority of them were collected during the winter when there are relatively high aerosol inputs from lower latitudes (Shaw, 1995). Within the full ONLi cloud group, we identified subsets of clouds present in clean background and aerosol-influenced conditions. Results were also compared with an internal subset of clouds where aerosol conditions and cloud properties could be ascertained with even higher confidence (i.e., those clouds that were Measured > One km

above the surface, Optically thin (COD < ~3), collected at Nighttime, predominantly Liquid, and from the Top-layer, henceforth referred to as "MOONLiT" clouds). The criteria for the cloud groups and aerosol classifications are summarized in Table 1. Justification for these criteria and descriptions of the individual datasets used for sample selection are described in more detail below.

### 2.1.1 CALIPSO

Aerosol vertical distribution, cloud top height, cloud base height, cloud optical depth, and initial approximate cloud phase were obtained from the polar-orbiting CALIPSO satellite lidar v. 3.01 level 2, 5-km aerosol profile and cloud layer products at 532 nm. These data have a vertical resolution of 30 m within layer (up to 8 km) where most predominantly liquid Arctic Ocean clouds were found. Before averaging, along-track cloud profile data were collected at a horizontal resolution of 1/3 km. Averaged aerosol data have a horizontal resolution of between 5-80 km, with the horizontal resolution increasing with

aerosol concentration. For example, in clear air with no detected aerosols, the horizontal resolution is 80 km; in strong aerosol layers, the horizontal resolution providing adequate signal-to-noise can be as low as 5 km (Vaughan et al., 2009).

Because our samples were taken at night, Moderate Resolution Imaging Spectroradiometer (MODIS) optical depths were not available. Instead, the CALIPSO product was used to measure CODs, as it offers substantially higher data availability in the optical thickness range of interest (COD < 3) than CloudSat (Christensen et al., 2013). Only non-quality-flagged (i.e., the

highest quality) CALIPSO COD data were used. CALIPSO cloud optical depth uncertainties rise with COD due to uncertainties in the lidar ratio in liquid clouds with COD > 1 (CALIPSO Quality Statements: Lidar Level 2 Cloud and Aerosol Layer Products, Version releases: 3.01, 3.02). We excluded COD data with uncertainties ≥ 75% of the COD value (these constituted ~5% of all cases).

Because it can be difficult to accurately separate Arctic aerosol from diamond dust and thin ice clouds using backscatter data

(*M. Vaughan*, pers. comm.; Grenier and Blanchet (2010)), we focused on CALIPSO liquid-containing clouds. To gain greater confidence in the aerosol classification within the MOONLiT subset, ice clouds were not allowed in those profiles. Note that CALIOP cloud "phase" indicates only whether the cloud predominantly contained liquid or ice; there is no mixed-phase designation. At a later step, CloudSat data were used to further refine cloud phase information.

CALIPSO data were obtained over the Arctic Ocean between 60-82°N and between 1 January 2008 – 7 December 2009

(during the latter part of CloudSat epoch 2). To obtain the lowest possible comparable detection limit, the analysis was restricted to nighttime clouds. Here, nighttime profiles are taken in the CALIPSO orbit over the hemisphere of Earth that is

dark at any given time, and so the borders of this hemisphere may include some low-light conditions. MOONLiT clouds were additionally restricted to upper-layer clouds only. We focused on ONLi clouds present between 0.2 and 8 km above the surface to enable better below-cloud aerosol detection. MOONLiT cloud cases were further restricted to above 1 km for better comparison to high-quality CloudSat data. Very few predominantly liquid clouds are expected above 8 km. Clouds

were included only when the feature's optical properties scored between 70 and 100 in the cloud-aerosol discrimination (CAD) algorithm (a high confidence cloud determination) (Liu et al., 2009). The lidar-determined presence of a below-cloud aerosol layer was a key criterion in identifying clean background clouds with confidence, as discussed further in Sect. 3.1. Thus, the analysis was limited to non-opaque clouds (COD < ~3), as determined by the 532 nm Extinction Quality Control flag.

The "clean, background" cloud subset met the above criteria, but no aerosol features were permitted above or below cloud, even when air masses had been horizontally averaged across 80-km resolution in the CALIPSO aerosol detection algorithm, which is the resolution that detects weak aerosol layers with highest confidence. Given these constraints, the backscatter aerosol detection limit for "clean background" clouds is as low as possible, and should have only negligible variations based

on detector noise and background molecular scattering and $O_3$ densities above cloud (Vaughan et al., 2009). Because CALIPSO cannot always detect dilute aerosols (Di Pierro et al., 2013; Kacenelenbogen et al., 2014; Rogers et al., 2014; Winker et al., 2013), particularly below-cloud where the lidar signal has been reduced, "clean background" clouds were also required to have modeled above and below-cloud FLEXPART ("FLEXible TRAjectory model", (Stohl et al., 1998, 2005)) black carbon concentrations of < 30 ng C m$^{-3}$ (see Sect. 2.1.3 and 3.1 for further discussion). The "aerosol-influenced" subset

had aerosols with CAD scores between -100 and -70 (high confidence aerosol classification) above or below the cloud and FLEXPART modeled below-cloud black carbon concentrations of > 30 ng C m$^{-3}$. The geographical distributions of the all-cloud, clean-cloud, and aerosol-influenced cloud sets are shown in Fig. 1.

### 2.1.2 CloudSat

CloudSat cloud profiling radar data are collected at a vertical resolution of 240 m. CloudSat has a wider swath than

CALIPSO (1.4x1.8 km) and it takes measurements on the same polar orbit, only seconds ahead of CALIPSO. Because the CloudSat radar does not accurately estimate cloud properties below ~0.7-1 km agl (Huang et al., 2012; Mioche et al., 2015). CloudSat data were provided only for clouds with bases ≥ 0.75 km agl. Some of the very thin clouds detected by CALIPSO had radar reflectivities that were too low to be detected by CloudSat, and CloudSat may sometimes mistakenly assign precipitating ice as a cloud (de Boer et al., 2008). Therefore, radar reflectivity data and CloudSat reflectivity-derived cloud

parameters, where available, were obtained from the height bins closest to where CALIPSO detected a cloud.

Average reflectivity between the CALIPSO-determined cloud top and base was obtained from the CloudSat 2B-GEOPROF version R04 dataset. Cloud phase and precipitation occurrence were acquired from 2B-CLDCLASS-LIDAR version R04

estimates (Wang, 2013). In this product, cloud phase is determined from a combination of CALIPSO water layer detection and integrated backscattering coefficient, temperature, CloudSat reflectivity, and an assumed temperature-dependent reflectivity threshold for ice particles (Zhang et al., 2010). This phase classification is uncertain for clouds with reflectivities of < -29 dBZ (the CloudSat sensitivity limit), and for very thin clouds due to the coarse vertical resolution of the instrument.

As we focused on cold, optically thin clouds in this study, many (~29%) of our samples were below the CloudSat detection limit. Thus, phase was only assessed in clouds with cloud phase certainty values of > 5 and with reflectivity values of > -29 dBZ. Infrequently, clouds that met the CALIPSO criterion in Table 1 were classified as predominantly ice phase by the 2B-CLDCLASS-LIDAR product; these cases were excluded from the analysis for simplicity, despite the potential for supercooled water to be misclassified as ice particles (Van Tricht et al., 2016).

Estimated mean liquid cloud droplet effective radii ($r_{el}$) were obtained from the CloudSat 2B-CWC-RO version R04 product (LO_RO_effective_radius) (Austin and Stephens, 2001). We chose this CloudSat $r_{el}$ product, which assumes that all particles are liquid, for two reasons: 1) CALIPSO had independently assigned the clouds a predominantly liquid phase, and 2) uncertainties in the other liquid $r_{el}$ data product available for nighttime samples (RO_liq_effective_radius) may be fairly high because of a reliance on an overly-simplistic, temperature-dependent phase partitioning scheme (e.g., de Boer et al. (2008);

Lee et al. (2010)). Where available, $r_{el}$ data were averaged over vertical regions within the CALIOP-determined "liquid" phase cloud base and top. Sometimes the corresponding CloudSat-determined cloud base and top were slightly different. In these cases, CALIOP heights were used because of its better ability to detect liquid droplets, and because CloudSat may sometimes misclassify precipitating ice as part of the cloud (de Boer et al., 2008), which can lead to overestimation of $r_{el}$. Quality-flagged data were excluded, such as observations from precipitating clouds, as determined from the CloudSat 2B-

CLDCLASS-LIDAR version R04 product. Note: although we counted the number of cases where precipitation occurred for comparison at a different step, precipitating cases were otherwise excluded from most other derived cloud parameters in the analysis. These cases were excluded in order to obtain comparable data across cloud characteristics, which was particularly important for the longwave emissions calculations detailed in Sect. 2.2 that included the $r_{el}$ as one of several input parameters.

We present some limited CloudSat-derived $r_{el}$ data here, but it is important to note the fairly high uncertainties in some of these data. Aside from the assumption of liquid phase, there is a known bug in the CloudSat code that might cause $r_{el}$ in liquid clouds to be overestimated, and to our knowledge there has been no extensive validation of the CloudSat 2B-CWC-RO $r_{el}$ product in the Arctic. de Boer et al. (2008) found fairly reasonable agreement, with perhaps some overestimation, between CloudSat-determined $r_{el}$ in mixed-phase clouds compared to $r_{el}$ measured from ground-based instruments. However,

only a few samples were collected with the in-cloud constraint in that study. The cumulative uncertainties in $r_{el}$ on the radiative impact results are discussed further in Sect. 3.5.

### 2.1.3 FLEXPART

The locations of combustion aerosol plumes were modeled using BC from the FLEXPART model (Stohl et al., 1998, 2005). The FLEXPART model has been used extensively to study pollution and smoke transport in the Arctic, and is well-validated for this purpose (Damoah et al., 2004; Eckhardt et al., 2015; Forster et al., 2001; Paris et al., 2009; Sodemann et al., 2011; Stohl et al., 2002, 2003, 2015). We chose BC as a combustion aerosol tracer because it represents aerosol removal better than a gaseous tracer like carbon monoxide, and because FLEXPART can largely capture the Arctic BC seasonal cycle (Eckhardt et al., 2015) that is driven by a combination of seasonal changes in emissions, atmospheric transport patterns and removal processes. In some cases, wildfires can emit large amounts of light absorbing organic carbon aerosols (or "brown carbon") without emitting large amounts of BC (e.g., Chakrabarty et al. (2016)). In these cases, FLEXPART BC may not represent smoke aerosols well.

For this study, as in Eckhardt et al. (2015), FLEXPART was driven with meteorological analysis data from the European Centre for Medium-Range Weather Forecasts (ECMWF) at a resolution of 1˚ longitude and 1˚ latitude. BC emissions were based on the ECLIPSE emission inventory (Stohl et al., 2015), which also includes emissions from gas flaring, and biomass burning emissions. In the model simulations, BC was removed from the atmosphere through dry deposition, and wet scavenging both below and within clouds. However, no transformation of BC from a hydrophobic to a hydrophilic state was considered and removal parameters were chosen as typical for a hydrophilic aerosol. FLEXPART-modeled BC concentrations were calculated for the years 2008 and 2009 at a horizontal resolution of $1^o$ latitude and $2^o$ longitude and at 0.05, 0.2, 1, 2, 3, 5, 7, and 10 km agl. Below-cloud BC concentrations were taken to be the closest modeled concentration available to 0.5 km below cloud base. When there were multi-layer clouds and the next cloud top was < 1 km away, the concentration closest to the middle distance between the two clouds was used instead.

### 2.2 Ancillary datasets

Aircraft out-of-cloud black carbon data were obtained from NASA's Arctic Research of the Composition of the Troposphere from Aircraft and Satellites (ARCTAS) campaign (Fuelberg et al., 2010; Jacob et al., 2010; Kondo et al., 2011). The aircraft data with highest aerosol particle concentrations were clustered between 50-60$^o$ N during this campaign. Thus, we included aircraft data from between 50-82$^o$ N (subarctic + Arctic) in order to assess comparable ranges of dilute and concentrated aerosols expected to be present over the Arctic**.** Submicron aerosol dry size distributions between 0.06–1 μm were measured from a DMT Ultra-High Sensitivity Aerosol Spectrometer (UHSAS) between 0-2.1 km (2.9 km for springtime samples). Submicron aerosol scattering data at 532 nm were obtained from a Radiance Research (RR) nephelometer and were corrected for truncation errors. Submicron aerosol scattering coefficients at 450 and 700 nm were estimated as the difference between total scattering from a TSI 3563 Integrating Nephelometer and the RR nephelometer when the fine mode aerosol fraction exceeded 0.6. Ambient total scattering coefficients at the three wavelengths were obtained from the TSI nephelometer, and were corrected for truncation errors following Anderson and Ogren (1998). Aerosol absorption

coefficients at 450, 532, and 700 nm were measured with a RR three-wavelength Particle Soot Absorption Photometer (PSAP).

An aircraft-derived, 180° backscatter coefficient is calculated following Sawamura et al. (2017) in order to compare the in situ data to that from CALIOP (units of $Mm^{-1}$ $sr^{-1}$). First, the measured dry, submicron aerosol size distribution, scattering coefficient, and absorption coefficient at 532 nm are input into a Mie theory model to determine the aerosol effective dry refractive index. Next, a hygroscopic growth factor was applied to the dry size distribution in the Mie theory model to reproduce observed humidified light scattering and thus derive the aerosol refractive index at ambient relative humidity. The 180° backscatter coefficient then follows from Mie theory using the adjusted size distribution and refractive index. This method is best suited for spherical particles, which we assume dominate the ARCTAS samples based on the main aerosol sources during the campaign (non-dust background aerosols, anthropogenic pollution and smoke (Jacob et al., 2010)).

Several other supplemental datasets were used for cloud environmental context. ETOPO1 Bedrock GMT4 data (Amante and Eakins, 2009) were used to identify cloud profiles over the Arctic Ocean region. NOAA/NSIDC Climate Data Record of Passive Microwave Sea Ice Concentration, version 2 data (Meier et al., 2013; Peng et al., 2013) were used to approximate the fractional sea ice cover over ocean at the specific month and location of each profile. A sample was classified as being primarily over sea ice or open ocean when the sea ice fraction at the given location and month was > 80% or < 20%, respectively.

Lastly, integrated surface longwave (4-30 μm) radiation was calculated with an updated Santa Barbara DISTORT Atmospheric Radiative Transfer program (SBDART, (Ricchiazzi et al., 1998)). Shortwave effects are not expected to be significant during nighttime conditions. Following McComiskey and Feingold (2008), the calculations assume homogeneous cloud cover and spectrally uniform surface albedo. Median surface longwave reflectivity (R) for open ocean and sea ice in clear conditions with no clouds or aerosols (0.64 and 0.69, respectively) was calculated from MERRA 2 output (GMAO, 2015) based on the times and locations of the data and the following formula (Josey, 2003):

(1) $$R = 1 - \frac{E - A}{I},$$

where E is the emitted longwave radiation from the surface, A is the net longwave flux into the surface from the atmosphere, and I is the downwelling longwave radiation from the atmosphere. Note: the A parameter above is proxied by the closest available parameter in the MERRA2 output, surface absorbed longwave radiation, and thus it does not include factors such as transmission, latent heat, or conduction and convection. Because even a 50% change in R would lead to < 1% error in the cloud longwave surface flux calculations, we expect the resulting uncertainty in R to have negligible impact on our results.

# 3 Results and Discussion

## 3.1 Correct identification of clean background conditions

To accurately characterize clean background conditions, it is necessary to detect combustion-related aerosol layers with confidence. For CALIPSO, dilute aerosols are least likely to be detected below-cloud due to signal attenuation inside the cloud (Di Pierro et al., 2013), but CALIOP can sometimes miss dilute aerosol layers even in clear air above clouds (Di Pierro et al., 2013; Kacenelenbogen et al., 2014; Rogers et al., 2014; Sheridan et al., 2012; Winker et al., 2013). Most previous works focused either on daytime samples, which have comparatively low signal-to-noise ratios, or on extinction data, which are more uncertain because they assume a prescribed lidar ratio. To begin quantifying the false negative rate relevant to this study, we used two independent methods to estimate the fraction of the time when nighttime Arctic CALIPSO data would not detect above-cloud aerosols when actually present.

First, we estimated the fraction of air masses containing various observed concentrations of aerosol tracers that would be detected at the reported theoretical 80 km resolution nighttime backscatter detection limit from Winker et al. (2009). This analysis is based on co-located aircraft backscatter, particle number, and BC data from the ARCTAS aircraft campaign (Fig. 2a). The results suggest that CALIOP would miss ~36% of slightly polluted air masses (i.e., BC concentrations > 30 ng m$^{-3}$) at 80 km resolution in nighttime air masses not below another feature. This estimate might be affected by errors from assuming Mie theory and a theoretical detection limit that may not be perfectly representative in the field, as well as errors caused by a limited amount of field data from scattered locations.

As an independent consistency check, we next determined the frequency at which aerosols were detected by both FLEXPART and CALIOP. To do so, we compared the fraction of observed clear sky (no-cloud) CALIOP profiles that were expected to contain aerosols at different simulated FLEXPART aerosol concentrations for January 2008 (Fig. 2b). These results suggested that CALIOP may not have detected up to ~33% of slightly polluted air masses (BC > 30 ng m$^{-3}$) above cloud, although this value likely overestimates the actual false negative rate given inherent model errors. This independent estimate is fairly similar to the previously estimated false negative rate, and so we expect the real-world above-cloud CALIOP false negative rate for dilute aerosols to be ~33-36%. Below-cloud errors would be higher, but are more difficult to quantify because of the variability of in-cloud attenuation.

Based on CALIPSO criteria alone, the above estimates suggest that aerosol detection uncertainties may be higher than desireable, particularly below cloud. We address this issue in two ways. First, we apply the criteria for determining clean background cloud that depend not only on aerosol-free CALIPSO profiles, but also on modeled above- and below-cloud BC concentrations of < 30 ng m$^{-3}$ (see Sect. 2.1.3). We expect the model aerosol-occurrence criterion to substantially improve the classification confidence because coincidences of false negatives in both the CALIOP data and the model are likely to be rare (they are most likely to occur in dilute aerosol conditions). As such, this method should correctly identify clean

background clouds much more frequently than 64-67% of the time. Unfortunately, further quantification in the classification confidence is difficult because both model accuracy and the degree of below-cloud lidar attenuation are variable in time and space. Secondly, we assess the MOONLiT cloud subset along with ONLi cloud results. MOONLiT clouds are a subset of ONLi clouds that, among other criteria meant to enhance certainty in aerosol layer identification, are in the top layer (see

Sect. 2.1 and Table 1 for more details). Trends in MOONLiT cloud results are mainly noted only if they are dissimilar to those in the larger ONLi cloud group, and are otherwise provided in the supplementary material. To our knowledge, the combined CALIPSO and model criteria used here allow the most confident classification of background conditions currently possible for remote sensing studies of the Arctic.

### 3.2 Notes on limitations imposed by the methods

In order to have greater confidence in quantifying the regional scale aerosol indirect effects, this study is limited to ONLi clouds and their MOONLiT cloud subset. It is important to emphasize that the ONLi cloud group is not representative of all Arctic clouds. During our study period, ONLi clouds were present in only 5.3% of all total comparable nighttime cloudy profiles over the Arctic Ocean ("comparable clouds" defined as having a satisfactory in-cloud CAD score of 70-100 and with cloud bases > 200 m to exclude fog). Liquid-dominated clouds tend to be found at lower altitudes than thicker opaque clouds

and thus may not always be identified in multi-layer clouds using CALIPSO. However, even though the actual prevalence of these clouds may be somewhat underestimated, it is clear that ONLi clouds represent just a small fraction of all Arctic clouds. Thus, we emphasize that the aerosol indirect responses described in this paper are not necessarily representative of Arctic clouds in general.

Moreover, the cloud-selection criteria imposed by our methods may induce some uncertainties in the analysis. For example,

due to the low COD constraint, it is possible that some fraction of the cloud subset influenced by aerosols may be selected from a different group of cloud types than some fraction of the clean background cloud subset. As an illustration, in a subarctic aircraft case study presented in Zamora et al. (2016) (see Appendix A for further details), cumulus clean background clouds with an observed cloud thickness of ~250 m had CODs of ~5. These clouds would have been too optically thick for the CALIOP lidar to penetrate. However, highly comparable nearby clouds in a smoke plume had CODs

of only ~2. The cloud-property differences were likely driven by the aerosol (Zamora et al., 2016). In this example, only the subset of clouds influenced by smoke aerosols would have met this study's COD criterion and not the clean background cloud counterparts. Median reductions in COD were fairly minor for aerosol-impacted clouds relative to background clouds, and were not significant over open ocean, and so we do not expect this effect to have a large impact on our study.

Similarly, any aerosol-driven phase changes that shifted clouds between predominantly ice- and liquid-containing clouds (e.g., Girard et al. (2013)) could have eliminated or added samples from/to our study, also potentially adding some bias to our results. These uncertainties are difficult to quantify, but are likely to be much smaller than the error that would be

introduced by expanding the dataset to include other, non-ONLi cloud subsets that would be characterized with greater uncertainty.

**3.3 ONLi cloud characteristics in clean marine background conditions**

In our study, sampled clouds were thin by definition and were thus unlikely to occur under very turbulent conditions. The range in turbulence covered in the sample set was also likely limited during polar night due to the lower variability in external heating and generally high static stability of the Arctic atmosphere. Nonetheless, we expect that clouds over the open ocean are impacted more by thermodynamic coupling with the surface (Shupe et al., 2013) than over sea ice, where surface-based inversions occur more frequently (Ganeshan and Wu, 2015). In this study, we stratify clouds into these two regimes, to distinguish the effects of systematic differences in atmospheric stability and large-scale atmospheric and surface forcing between the two systems (Curry et al., 1996; Jaiser et al., 2012; Taylor et al., 2015).

ONLi clouds were more likely to overlay another cloud layer over open ocean than over sea ice, as demonstrated by the average height of the next below-cloud feature (Fig. 3b, Table 2). A similar result was also observed previously at the SHEBA ship-based observatory (Intrieri et al., 2002). There are also differences between shallow and higher clouds. Shallow clouds are defined here as having cloud bases < 1.1 km asl, based on the lower quartile range of the cloud base height data. Over both open ocean and sea ice, shallow clouds are warmer and are more likely to have a liquid- vs. mixed-phase CloudSat designation (Tables S1 and S2). Shallow clouds are on average optically thicker, but geometrically thinner, than higher clouds. They are also less likely to be observed in multi-layer cloud conditions in both regimes ($p < 0.05$, permutation test), which may be due in part because they are systematically less observable due to lidar attenuation in higher thick cloud layers.

It is possible that some of the differences between shallow and high ONLi clouds are due to differences in cloud formation mechanisms. For example, previous studies suggest that shallow liquid-containing Arctic clouds might form from the advection of warm, moist air over a cool surface, whereas higher liquid-containing clouds might form from a longwave radiative flux divergence (Smith and Kao, 1995) or partial dissolution of a higher-level stratus cloud (Herman and Goody (1976). One previous model sensitivity study linked shallow liquid-containing clouds in a 3-day Arctic multi-layer cloud system with surface turbulent heat fluxes, and overlying liquid-containing clouds with large-scale advection and maintenance by radiative cooling (Luo et al., 2008). Because of these differences, shallow ONLi clouds were characterized separately in later analysis in order to better understand the influence of confounding meteorological factors on the results.

The different probabilities of cloud-layering occurrence over sea ice vs. open ocean and in cloud properties over different heights complicates comparisons between the two regimes. However, comparing only single-layer clouds with bases above 1.1 km, the median cloud base height of open ocean clouds is ~240 m higher (~480 m for MOONLiT clouds) than for clouds over the sea ice ($p < 0.05$, permutation test). Autumn ship-based cloud observations in the Chukchi and Beaufort Seas also

show higher cloud bases over the open ocean (Sato et al., 2012; Young et al., 2016). Over sea ice, the lower cloud heights and the presence of fewer multi-layer ONLi clouds compared to the open ocean (Table 2) are likely related to the lower height and greater frequency of surface-based inversions over Arctic sea ice, which can reduce surface moisture fluxes to higher altitudes (Bradley et al., 1992; Ganeshan and Wu, 2015; Zhang et al., 2011). Below 1.1 km, cloud base heights for
single-layer clouds are not significantly different between regimes.

Over the open ocean, clouds were also warmer than over sea ice, and a higher fraction of ONLi clouds were observed with very low layer mean reflectivity ($Z_m$), defined as $Z_m < -29$ dBZ (the CloudSat detection limit) (Table 2). The very low $Z_m$ clouds are geometrically and optically very thin (Table 2). Previously observed relationships between $Z_m$ and $r_{el}$ suggest that the very low $Z_m$ clouds also likely have smaller $r_{el}$ values (Frisch et al., 2002).

Because reflectivity was fairly low within the thin, predominantly liquid cloud profiles that fit our criteria, and temperatures were generally between -1 to -28 $^{o}$C, in many cases it was difficult to know for certain which clouds were of mixed vs. liquid phase. Of the clouds that were assigned a high-confidence phase classification by CloudSat, most contained some ice particles (93%, n=5238 for sea ice, and 79%, n=2992 for open ocean). We believe it likely that a comparatively higher fraction of the very low $Z_m$ clouds were present in the liquid-only phase. First, these clouds had very low $Z_m$ values
(indicative of small particles), and at the same time they were independently assigned a predominantly liquid phase by CALIPSO. Secondly, their median temperatures were warmer than clouds with higher $Z_m$ (by ~1-3$^{o}$C over sea ice, and nearly 1-7$^{o}$C over comparable altitudes over open ocean, Table 2). Relatedly, low $Z_m$ clouds were more than two times more likely to be found over open ocean than over sea ice (Table 2). Further study would be needed to fully verify phase for this cloud subset, but the indications that these clouds have higher liquid fractions are consistent with the observations that a)
Arctic liquid clouds tend to have smaller $r_{el}$ values than mixed-phase clouds (Hobbs and Rangno, 1985; Lance et al., 2011; Lebo et al., 2008; Rangno and Hobbs, 2001), and b) clouds over the open ocean (which were more likely to have very low $Z_m$ values (Fig. 4 a,d)) are also more likely to be liquid-containing (Cesana et al., 2012).

### 3.4 Aerosol impacts on clouds over sea ice

We expect that the greater uniformity in surface and meteorological conditions over sea ice will increase the likelihood of
being able to isolate aerosol impacts from meteorological noise, compared to the situation over the open ocean, and cloud characteristics were indeed fairly uniform over sea ice. We observed only minor differences in cloud base height between ONLi clouds present in clean background conditions and all ONLi clouds (Table 2). Above 1.1 km, the cloud base temperatures in clean background conditions were not significantly different from those in all air mass conditions. Below 1.1 km, clean background clouds appear to be found in slightly warmer conditions (by ~2 $^{o}$C) (Table S1).

Clean background clouds were significantly more likely to be precipitating than other clouds in both height bins (Table 2). This observation falls in line with aerosol-driven reductions in snowfall that have been predicted and observed previously, inside and outside of the Arctic (Albrecht, 1989; Borys et al., 2000, 2003; Girard et al., 2005; Lance et al., 2011; Lohmann et al., 2003; Mauritsen et al., 2011; Morrison et al., 2008). These observed reductions in precipitation are inconsistent with the glaciation indirect effect, in which ice formation would be expected to increase due to higher concentrations of combustion-related INP (Lohmann and Feichter, 2005). The presence of aerosols is also correlated with a significant reduction in radar reflectivity, generally associated with smaller particles on theoretical grounds (Fig. 4, Table 2). Correspondingly, there is also a significantly higher probability that clean background clouds detected by CALIPSO would also be detected by CloudSat than in all clouds or in aerosol-impacted clouds (Table 2).

The $r_{el}$ values are derived from radar reflectivity, and as such, aerosol-related decreases in reflectivity suggest smaller $r_{el}$ values. This observation follows expectations based on the Twomey effect, whereby aerosol particles acting as CCN create more droplets with smaller sizes, and is in line with previous studies in the Arctic that have observed smaller $r_{el}$ correlated with increasing influence of aerosols (Coopman et al., 2016; Lubin and Vogelmann, 2006; Peng et al., 2002; Tietze et al., 2011; Zamora et al., 2016; Zhao and Garrett, 2015). Here, non-shallow clouds > 1.1 km were associated with a systematic decrease in the cloud droplet effective radius as expected aerosol influence rose, and the estimated mode $r_{el}$ was respectively 10.3, 10.1, and 9.8 μm for the ONLi clean cloud, all cloud, and the aerosol-influenced cloud subsets. This reduction was similar in the MOONLiT subset, at 10.5, 10.3, and 10.0 μm, respectively (Table S3). Unfortunately, the differences in $r_{el}$ are available only for the thicker clouds that CloudSat was able to observe, and in some cases, data were available only for the middle sections of clouds, which are expected to have higher relative $r_{el}$ values. Thus, the estimated mean $r_{el}$ values presented here might be skewed higher than would be derived from a dataset that more fully sampled the cloud fields, and the differences compared to clean background cases could underestimate actual differences. The difference in estimated ONLi $r_{el}$ is about half of a previously reported, regionally integrated value for all Arctic clouds. Using MODIS $r_{el}$ estimates in thicker clouds (median COD ~ 11) with temperatures between 0-2 $^{o}$C, Tietze et al. (2011) saw an ~1 μm difference between the very cleanest clouds and median clouds. Note that these regionally averaged net changes in $r_{el}$ are much smaller than would be expected locally in very polluted clouds (e.g., Zamora et al. (2016)). Also note that decreases in $r_{el}$ are not significant in shallow clouds (Table 2). We hypothesize that shallow ONLi clouds may be subject to different meteorological forcing than non-shallow clouds >1.1 km, as discussed in section 3.3, and that this forcing might overwhelm cloud sensitivity to aerosols.

There are differences between cloud thicknesses in clean background air and other air masses that suggest the potential for meteorological co-variability in the samples. Clean ONLi clouds are optically and geometrically thinner than the other cloud groups (Table 2). Lower moisture associated with continental airflow that carries the aerosol might explain this difference (Lohmann and Feichter, 2005), if recent surface contact with warmer mostly mid-latitude regions did not enhance moisture. However, in two related remote sensing studies where Arctic clouds were tightly binned within related meteorological

groups, COD differences still appeared, and thus the authors attributed these differences to aerosol-driven changes in liquid water path (LWP) (Coopman et al., 2016; Tietze et al., 2011).

We also observed a small but significant increase in the portion of detected liquid phase clouds within sea ice clouds above
1.1 km (Tables 2 and S1). The trend in phase was not significant in MOONLiT cases (Table S3), and as with $r_{el,}$ it was also not significant in shallow clouds (Table 2). However, only a strong trend in MOONiT cases would be significant due to the very small sample size, and differential meteorological forcing on shallow clouds might overwhelm cloud sensitivity to aerosols at lower altitudes.

It is difficult to say whether the aerosol-related impacts on precipitation and radar reflectivity observed here are simply indicative of Twomey effects on liquid droplets, or whether some more complex mixed-phase and/or meteorological dynamics are also involved. One previous aircraft-based study offered some evidence to suggest that Twomey effects on droplet size may reduce the efficiency of secondary ice formation in the Arctic, particularly for thin clouds (Jackson et al., 2012), which would be consistent with the greater fraction of clouds estimated as liquid phase in non-background clouds.
However, low sample number and surface/ meteorological variability make this mechanism difficult to conclusively demonstrate on a larger scale. Laboratory studies indicate that smaller droplets may also lower the probability of critical ice embryo formation (Pruppacher and Klett (2010)).

The "deactivation effect," whereby sulfates reduce ice nucleating particle efficiency (Du et al., 2011; Girard et al., 2005,
2013; Lohmann, 2017), could also be consistent with our observations. Some limited in situ data support the occurrence of this mechanism (Jouan et al., 2012), but remote sensing data are contradictory (Grenier et al., 2009; Grenier and Blanchet, 2010), perhaps in part because of high uncertainties in below-cloud aerosols, and a focus on ice phase clouds, where it is more difficult for CALIPSO to accurately separate aerosols from ice particles. Other possible mechanisms that could explain the observed aerosol-related impacts on cloud properties are that polluted air might contain fewer ice nucleating particles
(INP) than clean background air (Borys, 1989), and/or that riming efficiency could be reduced (Lohmann and Feichter, 2005). If the very low $Z_m$ ONLi clouds in our study do indeed contain fewer cases with ice particles as we suspect (see Sect. 3.3 above), the greater presence of very low $Z_m$ clouds in aerosol-influenced conditions (Fig. 4) would support the possibility of these mechanisms dominating within the ONLi cloud subset. As more information is needed to verify phase in very low $Z_m$ clouds, for now this possibility remains conjecture.

### 3.5 Aerosol impacts on clouds over the open ocean

Whereas cloud properties over sea ice were relatively tightly constrained, there was a much larger range in cloud properties over the open ocean (Table 2) that may in part reflect the greater variability and higher magnitudes of surface turbulent heat

and moisture fluxes over open ocean (e.g., Morrison et al. (2008); Strunin et al. (1997); Taylor et al. (2015)). Variability reduced our ability to compare clouds within this regime, as did the uneven vertical distribution of aerosols. CALIPSO-detected aerosols in the Arctic are most frequently found at altitudes below 2 km (Devasthale et al., 2011b; Di Pierro et al., 2013; Kafle and Coulter, 2013; Winker et al., 2013). Over the open ocean, the median ONLi cloud base was above this level

(2.1 km), and the median cloud base in the clean background cloud subset was even higher (2.6 km). The greater likelihood of clean background clouds being found at higher altitudes than non-background clouds likely induces a categorical bias in the cloud properties shown in Table 2.

To better understand any meteorological bias induced by aerosol height differences between clean vs. non-clean clouds, but

still retain a sample size from our 2-year dataset that is as informative as possible, we separated clouds found over open ocean into three cloud-base-height bins (Table S2), and summarized the resulting information in Table 2. As over sea ice, the first bin includes clouds with base heights between 0.2-1.1 km. This range encompasses the lower quartile range of all open ocean clouds, isolating the shallow clouds that were observed to have different characteristics from the higher clouds in clean background conditions (Sect. 3.3). This range also happens to coincide with the lower quartile range of sea ice clouds

so that these two bins are more or less comparable to each other with respect to cloud-base height. The second bin covers 1.2-3.2 km (the interquartile range of open ocean clouds). The last bin includes clouds with bases > 3.2 km. Although aerosol-influenced clouds still appear most often near the bases of their bins, the median cloud height and temperature differences within bins are fairly small (Table S2). Altitude-related biases from aerosol vertical distributions can be one cause for the loss significant trends over the open ocean with altitude binning, indicated by the blue coloring in Table 2. A

loss of significance might also be caused by differences in cloud-aerosol response with altitude, as is observed in shallow and non-shallow clouds over sea ice (section 3.4); the general reduction in sampling when the data are stratified could also be a contributing factor.

There are some significant differences between clouds with and without aerosol influence in non-shallow ONLi clouds with bases above 1.1 km. Similarly to clouds over sea ice, radar reflectivity is reduced with higher aerosol influence, and the

fraction of low $Z_m$ clouds increases (Table 2). Median $r_{el}$ dropped by 0.4 μm in aerosol-impacted cases vs. clean background cases, compared to a 0.5 μm reduction over sea ice. Clouds with bases > 1.1 km, especially those at higher altitudes, are also thinner.

The reflectivity and $r_{el}$ trends were not consistently observed in the MOONLiT subset, likely because smaller sample size caused the lack of statistical confidence in the binned samples (see Table S3). However, in a similar study using MODIS

data for liquid clouds over the Arctic, Coopman et al. (2016) found significant trends in $r_{el}$ with greater predicted aerosol concentrations when they stratified their results by lower tropospheric stability (LTS), which is much greater over sea ice than over open ocean (Taylor et al., 2015). Like us, they found that the trends were weaker for regions with less expected

LTS (which in our case would be over open ocean). The MOONLiT subset also had a significantly greater fraction of clouds that were assigned a liquid phase in aerosol-influenced samples compared to clean background samples for clouds where high quality CloudSat phase information was available above 3.2 km. This trend was not observed in the ONLi cloud subset, potentially because the differences between clean and aerosol-influenced cases were more ambiguous than in the MOONLiT cloud subset, but the trend toward more liquid clouds in aerosol-influenced conditions was also observed in the higher ONLi cloud bin over sea ice. It is unclear whether a similar trend in phase would remain if more of the samples had contained high-quality phase data, so we can only remark that the association between aerosols and liquid phase clouds is not inconsistent with the deactivation effect or with reduced ice formation efficiency related to Twomey effects on droplet sizes.

In contrast to clouds found at higher levels, there were not many significant differences associated with aerosol-influence in shallow ONLi clouds with bases below 1.1 km. Moreover, some of the differences that were significant were small enough to not be very meaningful (e.g., a 20 m reduction in mean cloud base height with a corresponding 0 m difference in median cloud base height for clean clouds compared to all ONLi clouds). This observation suggests that dynamics might be overwhelming any aerosol changes to cloud microphysics in this regime, although our sample size for CloudSat derived parameters was reduced by only assessing those clouds that were > 750 m above the surface to avoid ground clutter of the instrument. Median cloud base heights in aerosol-influenced clouds were slightly higher (120 m) than clean clouds, which might have contributed to slightly colder cloud top heights.

### 3.6 Upper bounds on regional surface radiative impacts

Over our two-year time period, we identified tens of thousands of predominantly liquid ONLi clouds over the Arctic Ocean (Table 2). This sample size and regional spread of the data are large enough that we make the assumption that the cloud characteristics provided in Table 2 approximate the net nighttime cloud characteristics that exist for this cloud subset after exposure to the full spectrum of environmental conditions in each regime (sea ice and open ocean). We calculated the *maximum regional* radiative impact of clean background ONLi clouds on the nighttime surface, based on the regional frequency of occurrence of observable ONLi clouds in nighttime profiles over the entire (cloudy or clear) Arctic Ocean during our time period (2.52% and 4.84% over sea ice and open ocean, respectively; 3.23% over the full Arctic Ocean domain). Table 2 clean background cloud characteristics were used to calculate longwave flux changes to the surface compared to clear air, assuming cloud homogeneity and a single cloud layer, estimated at 56.05-58.44 W m$^{-2}$ and 20.86-21.48 W m$^{-2}$ for sea ice and open ocean regions, respectively. Maximum regional radiative impacts were estimated by multiplying these longwave fluxes by the ONLi cloud regional frequency of occurrence. Note that the presence of lower-level clouds will reduce the regional impact of ONLi clouds on the surface. Variable input parameters for the radiative impact calculations included cloud base height, cloud thickness and COD, and $r_{el}$ for clouds over sea ice and open ocean. Parameter values were taken from Table 2 median values, except for $r_{el}$, where the interquartile range was used to reflect the

larger uncertainty in that parameter.

The estimated maximum regional radiative impact of clean background ONLi clouds during polar night was between 1.41-1.47 W m$^{-2}$ over sea ice and 1.01-1.04 W m$^{-2}$ over open ocean. Maximum regional ONLi cloud impacts on the surface were smaller over the open ocean in part due to lower cloud temperatures associated with higher median cloud altitudes (an effect also seen during the SHEBA campaign (Shupe and Intrieri, 2004)). This effect occurred despite there being more ONLi cloud cover over open ocean than over sea ice (a general trend that is also observed in total cloud fraction (Kay and L'Ecuyer, 2013)). Also, the higher open ocean clouds are expected to have lower liquid water paths (based on thinner CODs, Table 2), which influences longwave cloud forcing in very thin clouds that are not opaque in the infrared (Turner, 2007). For reference, using the CloudSat 2B-FLEXHR-LIDAR product, Kay and L'Ecuyer (2013) estimated the annual mean longwave forcing at the surface due to all clouds over sea ice and open ocean to be ~24-36 and 32-56 W m$^{-2}$, respectively, depending on location. Barton et al. (2014) model-mean estimates for cloud impacts on surface longwave downwelling radiation during polar night over sea ice above 70 $^{o}$N (within the 95% confidence interval for surface temperatures) were ~15-30 W m$^{-2}$. These published estimates included the impacts of non-ONLi clouds, which the current study does not.

We also estimated the maximum regional surface indirect radiative effect of aerosols on ONLi clouds over sea ice. To do so, we subtracted the maximum regional surface radiative impacts of the clean background cloud subset from the impacts expected of all observed ONLi clouds. Radiative calculations were not made for aerosol-driven effects on ONLi clouds over the open ocean due to the lack of significant differences in most relevant parameters and the altitude-based bias in the full open ocean dataset. As with background clouds, aerosol-indirect radiative effect estimates were made using the median cloud base and top heights, the median COD, and the $r_{el}$ interquartile range for sea ice clouds presented in Table 2. Based on this information, we estimate that excluding changes in cloud fraction, aerosols could have indirectly decreased current-day surface downwelling longwave fluxes during polar night over sea ice, from ONLi clouds specifically, by no more than 0.11 W m$^{-2}$ (~10% of the clean background effect), integrated over sea ice across the Arctic and for all aerosol concentrations. As with the background cloud estimates, this spatially integrated estimate assumes single layer cloud conditions. Estimated regional aerosol indirect impacts specifically from the shallow (base height < 1.1 km) sea ice ONLi clouds accounted for about half of this effect. In this instance, holding all other variables equal, aerosol-related changes in cloud optical depth were an order of magnitude more important for radiative effects than the changes in cloud droplet effective radius, and the changes in geometric thickness had nearly no impact on the longwave impacts. It is important to note that because this range is spatially integrated across the Arctic, local aerosol impacts in strong haze layers can be much higher (e.g., Garrett et al. (2004); Carrió et al. (2005); Zhao and Garrett (2015)). For example, Zhao and Garrett (2015) found that the local cloud indirect longwave forcing in single-layer stratus clouds at Barrow, Alaska in the upper quartile of combustion aerosol concentrations was 8.1-9.9 W m$^{-2}$ greater than in clouds associated with the lower quartile of combustion aerosol concentrations. In a similar study at Barrow, Lubin and Vogelmann (2006) used the lower and upper quartile of aerosol

particle concentrations to show that downwelling flux for high CN cases was 3.4 W m$^{-2}$ higher than for low CN cases.

To be clear, in estimating mean aerosol indirect effects in this section, we did not isolate absolute or local indirect aerosol effects from the confounding effects of meteorology and meteorological co-variability. Instead, we estimated the current-day impact of combustion-derived aerosols on the regional indirect effect that ultimately influences the current-day surface radiation (which includes any meteorological co-variability present during these two years). This study was limited to only two years of data; future studies with more data might provide a better representation of the full range in aerosol and meteorological conditions the Arctic experiences over longer timescales.

As a final note, in this study we did not account for any aerosol-driven changes in cloud fraction. Aerosol-driven changes in cloud fraction may have occurred, given the reduced precipitation and the shift in CloudSat-estimated cloud type from predominantly altocumulus to predominantly stratocumulus in increasingly aerosol-impacted conditions over sea ice (Table 2). If aerosols do increase cloud fraction, this effect could be the most important indirect impact that aerosols have on the Arctic's surface radiation budget, because the presence of cloud where there otherwise would not be one has more of a local impact on surface radiation than does a change to a cloud that is already present (Feingold et al., 2016; Sedlar and Devasthale, 2012; Shupe and Intrieri, 2004). Addressing these issues will require further study with additional types of data.

**4 Summary and Conclusions**

Aerosol indirect effects have uncertain, but potentially large, impacts on the Arctic Ocean surface energy budget. As a step toward reducing uncertainty in current-day aerosol regional indirect effects on the surface, here we have better constrained the characteristics of a small subset of clean, average and aerosol-impacted clouds for which we have relatively strong constraints on cloud properties and the associated aerosol environment. We focused on optically thin (COD<~3), predominantly liquid clouds collected at nighttime, which we termed "ONLi" clouds; they cover about 3% of the nighttime Arctic Ocean (5% of total non-fog cloudy regions). However, within the ONLi cloud subset, it was possible to gain a high confidence in classification of clean background conditions with existing satellite remote sensing data. Using combined CALIPSO, CloudSat, and model output, we identify clean background clouds with a frequency that is much better than 64-67% of the time for top-layer clouds. Although the exact frequency of confident identification of clean background conditions beyond this range is difficult to quantify, particularly for clouds beneath another cloud layer, the level of confidence in clean background classification represents a substantial improvement compared to any previous remote sensing study of the Arctic region, as best we know.

Within the ONLi cloud subset, we observed clear differences between clouds over open ocean and over sea ice, consistent with different surface and meteorological conditions in these two regimes. For example, when the surface is open ocean compared to sea ice, ONLi clouds are much more likely to overlay another cloud and to be present in liquid phase. A greater

frequency of multi-layer clouds over the open ocean might affect the retreat of sea ice, and in turn, how this changes the impact of clouds on surface radiation of the Arctic Ocean. However, further study is needed to expand this observation beyond just conditions that contain ONLi clouds. There were also noticeable differences between shallow ONLi clouds (cloud bases < 1.1 km) and higher ONLi clouds.  As expected, shallow clouds were warmer and more likely be assigned a

liquid- rather than mixed-phase CloudSat designation; they were also optically thicker and geometrically thinner.  These differences in cloud properties may be in part to due the differing cloud formation mechanisms for shallow clouds. Previous studies support this hypothesis (e.g., Herman and Goody (1976); Smith and Kao (1995); Luo et al. (2008)), as does the observation, from the present study, that shallow ONLi clouds are less sensitive to aerosols.

Except in shallow, open ocean clouds, we observed that ONLi clouds are susceptible to aerosols. Consistent with other studies, the presence of aerosols exceeding background levels in clouds over sea ice is associated with reductions in $r_{el}$, cloud geometric and optical thickness, precipitation, radar reflectivity, and COD. Perhaps due to greater boundary layer turbulent fluxes, clouds over the open ocean appear to be less susceptible to the influence of aerosols, although some changes in phase and thickness were observed in the altitude-binned samples presented here. Due to aerosol-induced ONLi cloud changes

over sea ice, we estimate that the region-wide maximum surface radiation impact during polar night is an ~0.11 W m$^{-2}$ cooling (~10% of the clean background cloud effect, excluding any impacts on cloud fraction which were not assessed here), with shallow clouds contributing about half of this signal. It is unclear from the current work what the impact over open ocean might be. The maximum region-wide direct radiative impact of clean ONLi clouds at night is estimated to be 1.0 W m$^{-2}$ and 1.4 W m$^{-2}$ over sea ice and open ocean regions, respectively. Note that the presence of multi-layer clouds and cloud

patchiness will reduce the radiative impact of ONLi clouds on the surface. Also, these maximum regional indirect effect estimates do not include any potential aerosol-driven changes in cloud extent, which could be important for estimating ONLi-cloud overall regional indirect effects. Thus, aerosol-driven changes in cloud fraction dominate the uncertainty in estimates of the overall indirect aerosol radiative impact on the nighttime Arctic surface energy balance, based on this method. Unfortunately, the cloud fraction over the Arctic Ocean is particularly difficult to constrain over short time scales

with passive remote sensing, given the low contrast between clouds and sea ice and long polar nighttime conditions, and due to very limited spatial coverage for active remote-sensing.

We find no evidence to suggest that the glaciation indirect effect is important within the ONLi cloud subset. Beyond that, we have no strong support for aerosol impacts on mixed-phase cloud dynamics, although we see some tantalizing evidence to

suggest that large liquid particles need be present for ice formation in non-shallow ONLi clouds. These findings are in line with and expand upon previous aircraft observations (Jackson et al., 2012), although the deactivation effect could also explain the results. Aerosols were associated with higher fractions of liquid phase clouds than in clean background cases in both sea ice ONLi clouds > 1.1 km and in open ocean MOONLiT clouds > 3.2 km (for which additional cloud-selection criteria were applied; Table 1), for cases when high quality phase data were available. Above 1.1 km, open ocean and sea ice

clouds influenced by aerosol were less reflective at 94-GHz. Where high quality CloudSat data were available, these clouds also had noticeably smaller estimated median $r_{el}$ values, which is in line with previous studies. Over sea ice, aerosol-influenced clouds were less likely to be precipitating. Moreover, the fraction of low $Z_m$ clouds increases with aerosol presence in both regimes and at all altitudes except in shallow open ocean clouds. These low $Z_m$ clouds are more likely to be

liquid-dominated, based on their lower radar reflectivity combined with their independently assigned, predominantly liquid phase designation by CALIPSO, their warmer median cloud temperatures, and relatedly, their > 2 times higher relative fraction over open ocean compared to sea ice. Together, these observations suggest that aerosols could play an important role in ice nucleation and nighttime radiative heating via possibly reduced ice formation efficiency related to Twomey effects on droplet sizes, or the deactivation effect on aerosol particles. However, more information on cloud phase in low-reflectivity

clouds is necessary to more fully explore these possibilities.

Although we limited this study to carefully describing average and clean background clouds within only a subset of remotely sensed Arctic Ocean clouds, we were able to provide a first observation-based estimate of regional scale aerosol indirect effects on the surface for such clouds, demonstrating one way in which remote sensing observations can be used to

quantitatively assess aerosol-cloud interactions on a regional scale in other conditions and locations as well. Given that so far only models have been able to estimate regional aerosol indirect effects on the surface energy balance, this study lays an important foundation for improving the quantification of aerosol indirect effects. The trade-off for selecting a small subset of clouds in this study is the low representativeness of ONLi clouds. To constrain observation-based aerosol impacts and nucleation processes on a larger scale for the Arctic Ocean, optically thick and ice-containing clouds must also be included.

Expanding this study to a longer time period would help better incorporate the natural variability in Arctic meteorology and aerosols that might not be represented during this 2-year period. Including daylit or summertime air masses would also be useful; mid-summer air masses tend to be cleaner than wintertime Arctic air masses and have a higher fraction of liquid-containing clouds (Van Tricht et al., 2016). Moreover, it would enable the use of MODIS data to examine cloud phase (e.g., via the DARDAR data product (Delanoë and Hogan, 2010)) and droplet distribution.

**Acknowledgements**

We recognize and thank the efforts and funding from the large number of people and agencies involved in making the following datasets available, including: the NASA Langley Research Center Atmospheric Science Data Center, which provided the CALIPSO data, the CloudSat Data Processing Center run by the Cooperative Institute for Research in the Atmosphere (CIRA), and the NASA ARCTAS program and members. Specifically we would like to thank Y. Kondo and B.

Anderson for making their ARCTAS data publically available. We also thank G. de Boer, J. Creamean, G. Feingold, K.B. Huebert, J. Limbacher, S. Platnick, A. Solomon, H. Telg, M. Vaughan, D.L. Wu, Y. Yang, H. Yu, and T.L. Yuan for helpful discussions. The research of L. Zamora was supported by the NASA ACMAP program, via an appointment to the NASA Postdoctoral Program at the NASA Goddard Space Flight Center, administered by Universities Space Research Association.

The work of R. Kahn is supported in part by NASA's Climate and Radiation Research and Analysis Program under H. Maring, and NASA's Atmospheric Composition Program under R. Eckman. The NILU team was supported by funding from NordForsk in the framework of eSTICC (eScience Tools for Investigating Climate Change at High Northern Latitudes).

**Appendix A**

In Zamora et al. (2016), the case study CODs were not presented. Here, we calculated the relevant CODs from the following relationship:

$$(2) \quad COD = \frac{3}{2} \frac{LWC \ (z_t - z_b)}{r_{el}},$$

where LWC is the liquid water content, $z_t$ and $z_b$ are cloud top and base height, respectively, and $r_{el}$ is the cloud droplet effective radius.

**Supplement link (to be provided by Copernicus)**

**Competing Interests:**

The authors declare that they have no conflict of interest.

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

**Table 1: Criteria used for cloud and air mass classification.**

| Data source type | ONLi clean clouds | ONLi all clouds | ONLi aerosol-influenced clouds | MOONLiT clean clouds | MOONLiT all clouds | MOONLiT aerosol-influenced clouds | Clear air |
|---|---|---|---|---|---|---|---|
| *CALIPSO v. 3.01 L2 532 nm aerosol profile data* | | | | | | | |
| Latitude: 60-82$^{o}$N | x | x | x | x | x | x | x |
| Nighttime | x | x | x | x | x | x | x |
| Uppermost cloud layer only | | | | x | x | x | |
| Cloud top altitude < 8 km asl | x | x | x | x | x | x | |
| Cloud base altitude > 0.2 km asl | x | x | x | | | | |
| Cloud base altitude > 1 km asl | | | | x | x | x | |
| COD < ~3 (no extinction QC flag) | x | x | x | x | x | x | |
| In-cloud CAD score between 70-100 | x | x | x | x | x | x | |
| CALIPSO "liquid"-phase only | x | x | x | x | x | x | |
| No cloud phase quality control flags | x | x | x | x | x | x | |
| No aerosol above cloud | x | | | x | | | |
| Aerosol observed above or below cloud | | | x | | | x | |
| No aerosol between cloud base and surface or next cloud top, whichever comes first | x | | | x | | | |
| Aerosol CAD score between -100 and -70 | | | x | | | x | |
| No clouds or aerosol anywhere in profile | | | | | | | x |
| No absolute profile CAD score values <70 | | | | x | x | x | |
| No ice allowed anywhere in profile | | | | x | x | x | |
| *FLEXPART model output* | | | | | | | |
| BC ≤ 30 ng C m$^{-3}$ | x | | | x | | | |
| BC ≥ 30 ng C m$^{-3}$ | | | x | | | x | |
| *CloudSat 2B-CLDCLASS-LIDAR data[a]* | | | | | | | |
| >750 m above ground | x | x | x | x | x | x | |
| Non precipitating clouds | x | x | x | x | x | x | |
| Liquid- or mixed-phase only | x | x | x | x | x | x | |
| Liquid-phase only (for $r_{el}$ measurements) | x | x | x | x | x | x | |

[a]As available for clouds with radar reflectivities above the detection limit of -29 dBZ

**Table 2: Median (interquartile range) and sample number (n) of Arctic Ocean ONLi cloud properties as classified by the criteria in Table 1, separated by reflectivity above and below detection limit (DL, -29 dBZ) and surface regime. Red (grey) color indicates significant (not significant) differences compared to clean background clouds, as determined at 95% confidence using a permutation test. Blue indicates that when binned by altitude[d], significance was lost[e]. . An asterisk indicates that the trend observed without binning was still observed in non-shallow clouds > 1.1 km (see text and Supplementary Tables 1 and 2 for more details).**

| Attribute | Zm | Sea ice | | | | | | Open ocean | | | | | |
|---|---|---|---|---|---|---|---|---|---|---|---|---|---|
| | | Background | n | All clouds | n | Aerosol-impacted[a] | n | Background | n | All clouds | n | Aerosol-impacted[a] | N |
| Base T (°C) | > DL | -18.9 (-21.8 to -16.0) | 5804 | -19.3 (-22.3 to -16.1) | 19504 | -19.3 (-22.9 to -14.8) | 800 | -13.2 (-18.7 to -7.8) | 3681 | -11.7 (-17.6 to -6.7) | 11339 | -13.8 (-18.6 to -8.7) | 487 |
| | < DL | -18.6 (-22.2 to -15.0) | 897 | -18.4 (-21.5 to -15.1) | 4594 | -18.5 (-22.1 to -15.1) | 391 | -8.4 (-17.0 to -3.4) | 1548 | -7.3 (-15.7 to -2.7) | 6206 | -9.8 (-17.0 to -4.9) | 346 |
| | All | -18.8 (-21.8 to -15.8) | 6975 | -19.1 (-22.2 to -15.8) | 25140 | -18.9 (-22.7 to -14.9) | 1261 | -11.7 (-18.2 to -6.0) | 5487 | -10.0 (-16.9 to -4.8) | 18499 | -12.3 (-17.7 to -6.6) | 879 |
| Top T (°C) | > DL | -23.6 (-27.4 to -20.1) | 5804 | -23.2 (-27.1 to -18.7)* | 19504 | -23.0 (-27.1 to -18.7)* | 800 | -20.2 (-25.8 to -13.0) | 3681 | -17.5 (-24.0 to -11.7) | 11339 | -20.0 (-24.8 to -14.4) | 487 |
| | < DL | -21.2 (-25.6 to -17.8) | 897 | -20.9 (-24.2 to -17.7)* | 4594 | -21.4 (-24.5 to -18.2) | 391 | -11.8 (-21.5 to -6.6) | 1548 | -10.7 (-19.6 to -6.0) | 6206 | -13.0 (-20.9 to -8.2) | 346 |
| | All | -23.3 (-27.2 to -19.6) | 6975 | -22.7 (-26.6 to -19.0)* | 25140 | -22.3 (-26.4 to -18.4)* | 1261 | -18.0 (-24.2 to -10.0) | 5487 | -15.1 (-22.4 to -8.6) | 18499 | -17.8 (-23.1 to -10.2) | 879 |
| Altitude, base (km) | > DL | 1.72 (1.30-2.38) | 5804 | 1.60 (1.12-2.20)* | 19504 | 1.78 (1.24-2.44) | 800 | 2.74 (1.36-3.70) | 3681 | 2.26 (1.18-3.40) | 11339 | 2.50 (1.60-3.40) | 487 |
| | < DL | 2.02 (1.42-2.86) | 897 | 1.78 (1.12-2.50)* | 4594 | 2.08 (1.54-2.68) | 391 | 2.32 (1.36-3.58) | 1548 | 2.02 (1.30-3.16)* | 6206 | 2.26 (1.54-2.98) | 346 |
| | All | 1.78 (1.30-2.38) | 6975 | 1.60 (1.12-2.26) | 25140 | 1.90 (1.30-2.56) | 1261 | 2.62 (1.36-3.64) | 5487 | 2.14 (1.18-3.82) | 18499 | 2.38 (1.54-3.22) | 879 |
| Thickness (km) | > DL | 0.96 (0.66-1.32) | 5804 | 0.78 (0.60-1.20) | 19504 | 0.72 (0.60-0.96)* | 800 | 0.84 (0.60-1.32) | 3681 | 0.78 (0.60-1.32)* | 11339 | 0.72 (0.60-1.11)* | 487 |
| | < DL | 0.60 (0.48-0.72) | 897 | 0.60 (0.48-0.72)* | 4594 | 0.54 (0.48-0.66)* | 391 | 0.06 (0.48-0.78) | 1548 | 0.06 (0.48-0.72) | 6206 | 0.06 (0.48-0.72) | 346 |
| | All | 0.84 (0.60-1.26) | 6975 | 0.72 (0.60-1.08) | 25140 | 0.66 (0.54-0.84) | 1261 | 0.72 (0.54-1.08) | 5487 | 0.66 (0.54-1.08)* | 18499 | 0.66 (0.54-0.84)* | 879 |
| COD | > DL | 1.14 (0.65-1.85) | 4160 | 1.00 (0.60-1.63) | 16234 | 0.84 (0.53-1.40) | 772 | 0.82 (0.39-1.54) | 3286 | 0.88 (0.44-1.51) | 10474 | 0.81 (0.48-1.26) | 463 |
| | < DL | 0.55 (0.30-1.11) | 816 | 0.63 (0.36-1.07) | 4372 | 0.53 (0.34-0.89) | 387 | 0.49 (0.23-1.09) | 1427 | 0.62 (0.29-1.21) | 5885 | 0.61 (0.33-1.12) | 339 |
| | All | 1.03 (0.55-1.72) | 5195 | 0.90 (0.52-1.51) | 21533 | 0.73 (0.42-1.15) | 1227 | 0.69 (0.29-1.41) | 4952 | 0.77 (0.35-1.40) | 17265 | 0.72 (0.37-1.18) | 847 |
| Multi-layer clouds | > DL | 75% | 5804 | 79% | 19504 | 91% | 800 | 90% | 3681 | 89% | 11339 | 94% | 487 |
| | < DL | 85% | 897 | 85% | 4594 | 95%* | 391 | 92% | 1548 | 91% | 6206 | 96% | 346 |
| | All | 77% | 6975 | 80% | 25140 | 92% | 1261 | 90% | 5487 | 90% | 18499 | 95% | 879 |
| BC at base (ng m$^{-3}$) | > DL | 15 (10-21) | 5804 | 26 (14-54) | 19504 | 60 (42-94) | 800 | 13 (8-20) | 3681 | 18 (10-36) | 11339 | 61 (42-95) | 487 |
| | < DL | 15 (11-21) | 897 | 24 (13-48) | 4594 | 54 (38-94) | 391 | 13 (8-19) | 1548 | 17 (9-36) | 6206 | 61 (40-105) | 346 |
| | All | 15 (10-21) | 6975 | 26 (14-52) | 25140 | 59 (41-94) | 1261 | 13 (8-19) | 5487 | 18 (10-37) | 18499 | 61 (41-102) | 879 |
| % < CloudSat DL[b] | All | 15% | 6194 | 21% | 21841 | 36% | 1163 | 33% | 4950 | 40%* | 16612 | 44%* | 850 |
| % Mixed-phase[b] | > DL | 95% | 4795 | 93%* | 15698 | 91%* | 681 | 79% | 2992 | 75%* | 9153 | 80% | 417 |
| % precipitating[b,c] | > DL | 18% | 5916 | 13% | 18125 | 11%* | 737 | 8% | 3283 | 8% | 10077 | 11% | 454 |
| $r_{el}$ (µm)[b] | > DL | 10.3 (9.4-11.2) | 4917 | 10.0 (9.2-11.0)* | 15414 | 9.8 (9.1-10.7)* | 650 | 10.0 (9.2-11.2) | 2729 | 10.0 (9.1-11.2) | 8420 | 9.7 (9.0-10.9)* | 368 |
| Reflectivity (dBZ)[b] | > DL | -20.4 (-24.3 to -16.7) | 5294 | -21.5 (-25.3 to -17.6) | 17287 | -22.8 (-26.4 to -18.8) | 745 | -21.7 (-25.7 to -16.9) | 3680 | -22.2 (-26.2 to -17.0)* | 11329 | -23.5 (-26.7 to -19.1)* | 487 |

[a]Aerosol-impacted, as determined in the third column of Table 1.

[b]For clouds with bases >750 m asl

[c]Precipitating clouds were included in this metric only; for all other attribute classifications, clouds were required to have no observed precipitation in order to be comparable with $r_{el}$ estimates that were most reliable in non-precipitating clouds.

[d]Samples were divided into altitude bins (< 1.1 and > 1.1 km over sea ice, and < 1.1, 1.1-3.2, and > 3.2 km over open ocean); significance was then assessed within each altitude bin, as with the non-binned data.

[e]Significance was presumed to be lost across altitude bins when there were multiple cases of non-significance among altitude bins or different trends in significance between altitude bins.

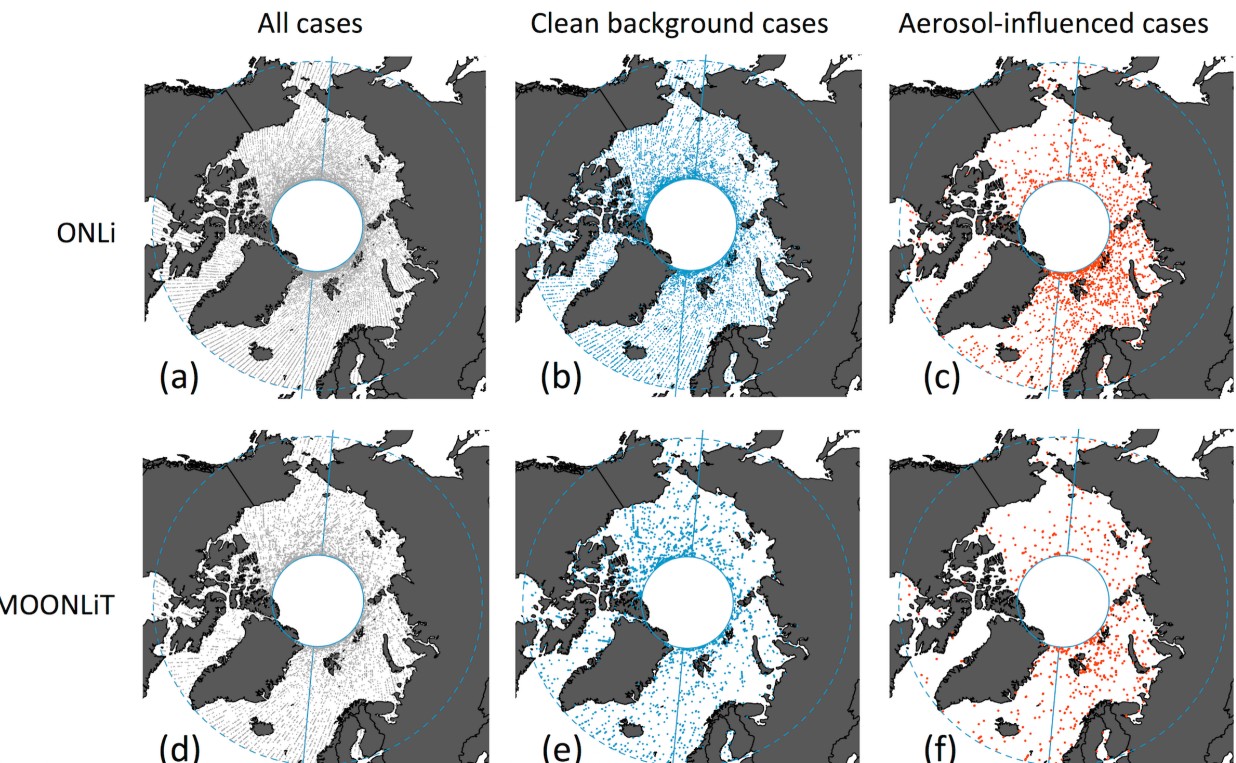

**Figure 1: The geographical distribution of ONLi and MOONLiT cloud profiles, where (a,d) grey indicates all cases, (b,e) blue indicates clean background cases, and (c,f) red indicates aerosol-influenced cases.**

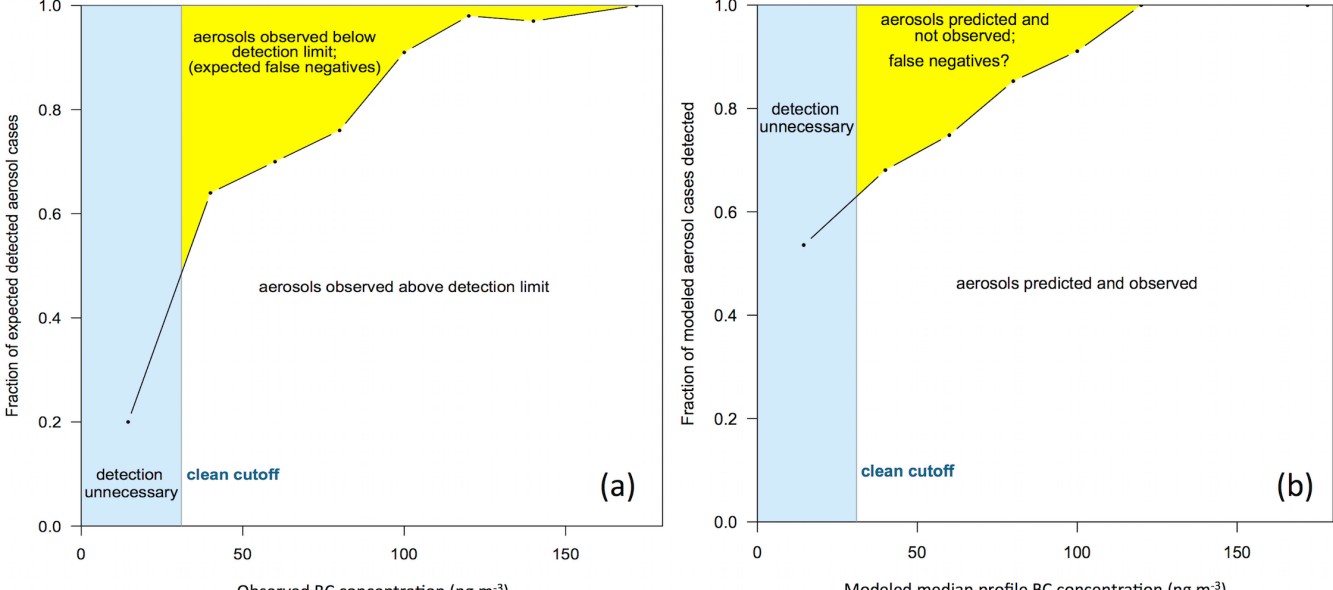

**Figure 2: Based on CALIPSO Arctic profiles under non-cloudy conditions, we compare a) the expected fraction and b) possible maximum fraction of false negatives (aerosol present but not detected) for the combustion tracer, black carbon (BC, ng C m$^{-3}$). The expected fraction of false negatives in panel a) was determined by comparing binned out-of-cloud 2008 ARCTAS-A and -B BC concentrations with the fraction of the total number of samples between 1-5 km that had converted backscatter values (Mm$^{-1}$ sr$^{-1}$) above the CALIPSO clear-sky nighttime backscatter detection limit from Winker et al. (2009) (see text for more details). Possible maximum false negative values in panel b) were determined by comparing the FLEXPART model's median BC concentrations between 0-10 km with the fraction of the total CALIPSO profiles under non-cloudy conditions during January, 2008 where aerosols were not detected. The clean cut-off below which air is taken as "clean" is assumed to be 30 ng BC m$^{-3}$.**

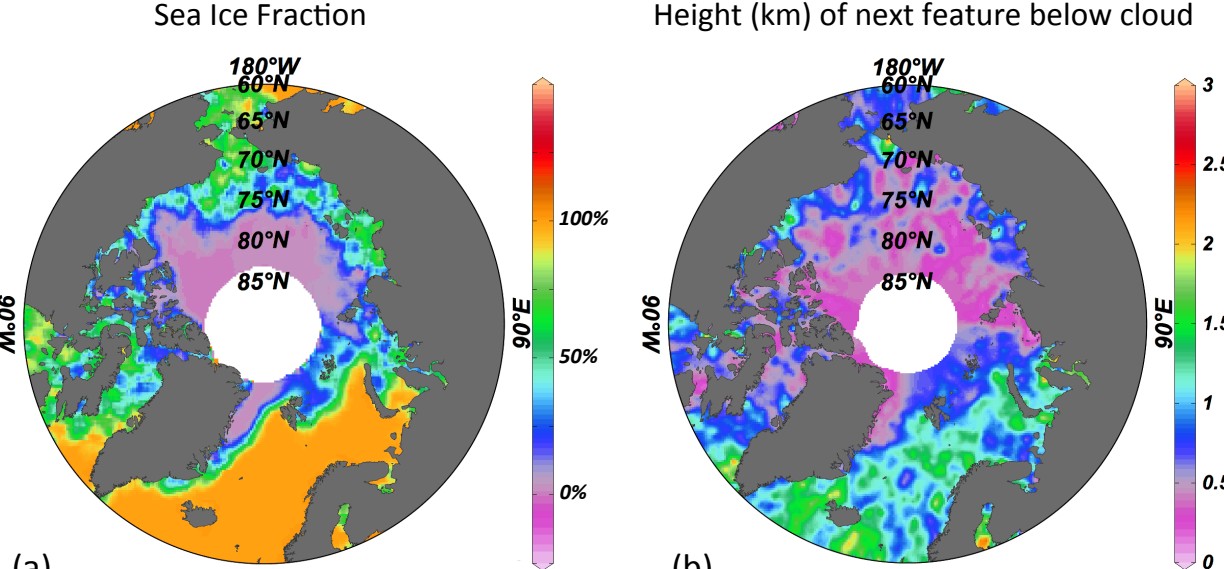

**Figure 3: The weighted-average gridded maps of features below individual cloud points from Fig. 1b for a) sea ice fraction, and b) height of the next lowest feature associated with individual cloud profiles, where a value of 0 indicates that the ocean surface or sea ice was the next lowest feature. Over open ocean, ONLi clouds were much more likely to overlay another cloud than over sea ice.**

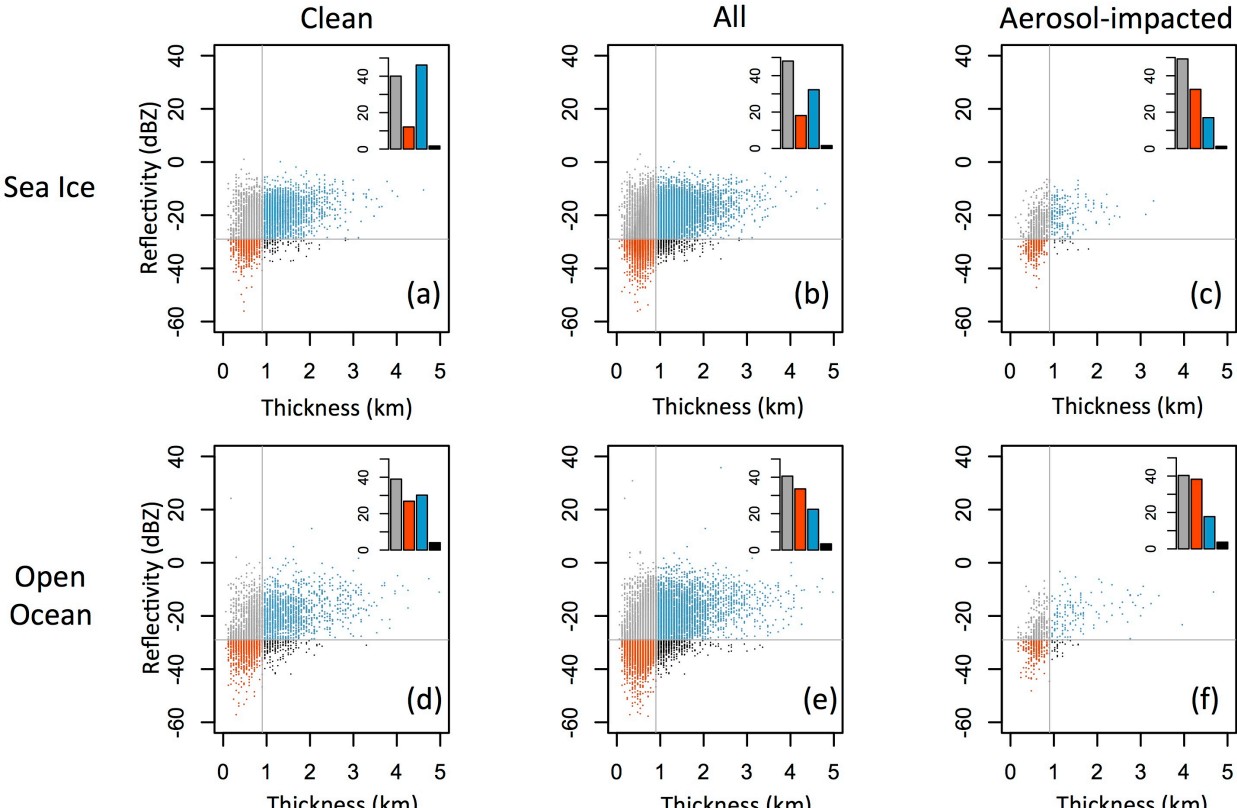

**Figure 4: A comparison of CALIPSO ONLi cloud thickness (km) with CloudSat reflectivity (dBZ), as separated by sea ice and open ocean regimes, and by clouds found in conditions labelled as clean background, all conditions, and aerosol-impacted conditions. To better show changes in the two parameters, plots have been divided into four quadrants (above (grey and blue) and below (orange and black) the CloudSat reflectivity detection limit of -29 dBZ), and above (blue and black) and below (orange and grey) a thickness of 0.9 km. In the upper right of each plot is shown the percent of cases within each quadrant, following the quadrant color scheme. Points represent clouds > 750 m asl.**