# Peer review of "Aerosol indirect effects on the nighttime Arctic Ocean surface from thin, predominantly liquid clouds"

_Atmospheric Chemistry and Physics, 2016_

## Referee Comment (RC1) · Anonymous Referee #1 · 31 Dec 2016

GENERAL COMMENTS:

Much of this analysis is interesting and relevant. Nonetheless I have some concerns with the paper as it is that primarily relate to the extremely intensive filtering of the data that has happened. The authors are left with only a handful of cases (It's not clear to me exactly how many, maybe thousands) from two years worth of satellite data. Generally the utility of satellite data is found in the large sample volume; an advantage eliminated in this study. Even after throwing out most of the data the authors then proceed to estimate the effect of the aerosol on cloud longwave radiation over the whole arctic. It is very difficult to believe that the handful of cases examined here can be representative of meteorological conditions over the entire arctic and throughout the year. It is not clear to me that this extensive filtering is necessary or even useful. In fact it may introduce undesirable sampling biases.

[Figure]

An example of this over-filtering of the data is eliminating clouds that are detected by CALIPSO but undetected by CloudSat. Many of these clouds will be shallow liquid-only clouds with small drop sizes (exactly the cloud type purported to be studied here) and yet they are thrown away. Wouldn't one interesting test be to determine if these clouds are more prevalent in the polluted conditions. This might be expected from the authors hypothesis. As another example what sense does it make to require the cloud optical depth to be less than 3. Once again don't we want to know if there are changes in the relative frequency of occurrence of these optically thick clouds in the presence of aerosol.

In short I found the justification for the filtering methodology to be lacking and I would really encourage either a convincing justification for why most of the data is thrown out or more appropriately just include include all cloud in the analysis.

Finally, the authors really want to get at the impact of aerosol on the cloud long-wave effect. The CloudSat data products (2B-FLXHR-LIDAR) have already calculated clear/cloudy fluxes for every pixel using combined input from CloudSat and CALIPSO cloud and aerosol profiles. The authors have put a good bit of work into identify clean and polluted conditions. All of this could be put together to simply calculate the aerosol effect for all cloud conditions without all of the filtering. Some more specific comments are below

SPECIFIC COMMENTS:

Section 2.1.2: By limiting analysis to cases where both CALIPSO and CloudSat identify approximately the same cloud height the authors throw out a great number of cases where clouds may have radar reflectivites below the detection threshold of the radar but are the thin liquid clouds of interest to the study. Eliminating clouds that have a base height greater than 1 km further aggravates this situation. In fact the authors have chosen a sampling strategy that minimizes the data availability from either instrument because it will be infrequent that clouds have optical depth less than 3 but still have

a radar reflectivity above the $\sim$ -28 dBZ CloudSat sensitivity. This is why it looks like there are maybe only a few hundred points on figure one. I can't reconcile this with the statement that 95% of the data are included in the analysis. How many pixels are included in the analysis? How many total pixels are there over the two year period?

Page 5, line 31: Why exclude precipitation cases? Don't we expect some aerosol influence on the occurrence of precipitation?

Section 2.3: The authors seem to recognize that the artificial filters that they are applying to the data may well introduce biases. So why not include all the clouds regardless of optical depth or detection by radar?

Fig 3: Where does sea ice data come from?

Fig 3. Does this map include only the filtered data points shown in Fig 1.

Page 11, Line 11: How is precipitation determined? Which product?

Page 11, Line 22: I see Fig 1 differently. To my eye there is a clear clustering of the data with substantially more aerosol cases north of Europe and relatively more clean cases north of Siberia and North America. This statement is not justified by the analysis.

Page 12, Line 8: It is fairly obvious that you won't find an optical thickness difference when you have artificially limited the range of optical thicknesses to less than 3.

---

## Referee Comment (RC2) · Anonymous Referee #2 · 25 Jan 2017

General comments: This manuscript by Zamora et al. presents an extensive study of thin liquid clouds over the Arctic and how these are affected by aerosol loading. The study combines satellite data from CALIPSO and CloudSat with FLEXPART modeling and aircraft measurements to better distinguish to which degree that the clouds were affected by aerosols. The study is limited to nighttime thin clouds between 1 and 8 km height and an estimation of the radiative impact of these clouds is provided. The manuscript is well written and contains detailed discussions regarding the uncertainties in the method and results. I recommend that the manuscript be published after answers to the following comments have been provided.

Specific comments: The study only includes nigttime clouds that have an COD < 3 and that are liquid. For the clouds to be included in the study they also must have an altitude between 1 and 8 km. In the methods section there are detailed descriptions of removal
of data due to several other criteria considering confidence in data etc. My question is how representative the clouds included in the study are for the general conditions in the Arctic. Could you provide an estimate of how common these liquid clouds are? If the clouds in this study represents the conditions during 80% of the time or 20% of the time makes a big difference. I believe that the second sentence in the abstract may be a bit bold if it turns out that these clouds are not very common in the Arctic.

The description of the data selections is very well written and detailed. However, it would be nice to know approximately how much data are lost at each step in the selection process.

Page 4, line 11: There are large land areas in parts of the described regions. Were these removed from the dataset?

Page 4, line 22: Were all the cases averaged to 80km resolution or do the different cases have different resolutions?

Page 7, line 9: Why is data 10 degrees further south than the satellite data included in the comparison?

Page 15, line 12: In the calculations of the indirect radiative effect of aerosols on MOONLiT clouds you write that you use the clean background cloud subset. Previously in the method you write that the parameters used in the calculations are cloud base height, cloud thickness and COD. For the cases over sea ice the COD is the same for the clean background and all cases datasets which means that the differences in the radiative effects is due to the difference in cloud base height (1.8 km vs. 1.9 km) and the difference in the cloud thickness (0.9 km vs 1.2 km). Did I understand this correctly? Could you comment on this?

Figure text figure 3: "where a value of 0 indicates that the ocean surface was the next lowest feature". Does ocean surface here also mean sea ice?

Technical corrections:

Page 12, line 7: optical thickness should be changed to COD.

---

## Short Comment (SC1) · 30 Jan 2017

This manuscript is very interesting and constitutes a unique contribution to the better understanding of aerosol indirect effect on Arctic clouds, given the key findings revealed by the combined CALIOP-CLOUDSAT data. I have one minor comments the authors can consider in the revision, which is as follows:

Page 2, lines 6-7: "3) the complexity of cloud responses to aerosol type and amount,": At least the following two papers can be cited to benefit the readers.

Chen T.M., Guo J.P., Z. Li, C. Zhao, H. Liu, M. Cribb, F. Wang, and J. He. A CloudSat perspective on the cloud climatology and its association with aerosol perturbation in the vertical over East China, J. Atmos. Sci., 73, 3599–3616, doi:10.1175/JAS-D-15-0309.1.2016.

[Figure]

Fan, J., Wang, Y., Rosenfeld, D., Liu, X.. Review of Aerosol–Cloud Interactions: Mechanisms, Significance, and Challenges. Journal of the Atmospheric Sciences 73, 11, 4221-4252,2016.

Interactive
comment

---

## Author Comment (AC1) · 13 Apr 2017

Zamora et al. ACPD Reviewer Responses

Response to reviewer #1

GENERAL COMMENTS:

Much of this analysis is interesting and relevant. Nonetheless I have some concerns with the paper as it is that primarily relate to the extremely intensive filtering of the data that has happened. The authors are left with only a handful of cases (It's not clear to me exactly how many, maybe thousands) from two years worth of satellite data. Generally the utility of satellite data is found in the large sample volume; an advantage eliminated in this study. Even after throwing out most of the data the authors then proceed to estimate the effect of the aerosol on cloud longwave radiation over the whole arctic. It is very difficult to believe that the handful of cases examined here can be representative of meteorological conditions over the entire arctic and throughout the year. It is not clear to me that this extensive filtering is necessary or even useful. In fact it may introduce undesirable sampling biases.

An example of this over-filtering of the data is eliminating clouds that are detected by CALIPSO but undetected by CloudSat. Many of these clouds will be shallow liquid-only clouds with small drop sizes (exactly the cloud type purported to be studied here) and yet they are thrown away. Wouldn't one interesting test be to determine if these clouds are more prevalent in the polluted conditions. This might be expected from the authors hypothesis. As another example what sense does it make to require the cloud optical depth to be less than 3. Once again don't we want to know if there are changes in the relative frequency of occurrence of these optically thick clouds in the presence of aerosol.

In short I found the justification for the filtering methodology to be lacking and I would really encourage either a convincing justification for why most of the data is thrown out or more appropriately just include all cloud in the analysis. Finally, the authors really want to get at the impact of aerosol on the cloud longwave effect. The CloudSat data products (2B-FLXHR-LIDAR) have already calculated clear/cloudy fluxes for every pixel using combined input from CloudSat and CALIPSO cloud and aerosol profiles. The authors have put a good bit of work into identify clean and polluted conditions. All of this could be put together to simply calculate the aerosol effect for all cloud conditions without all of the filtering. Some more specific comments are below.

We thank the reviewer for their helpful comments on our manuscript, which have improved the paper.

1) In the general comments above, the reviewer suggested that we provide, "either a convincing justification for why most of the data is thrown out or more appropriately just include all clouds in the analysis." We have addressed this suggestion/concern in three ways:

i.   At the request of the reviewer, we now have expanded our analysis to compare MOONLiT clouds with the broader group that they best represent: all nighttime optically thin, predominantly liquid clouds. Within this subgroup of clouds we still have fairly high confidence in the aerosol conditions surrounding the clouds. The nighttime criterion is kept to simplify the assessment of indirect effects (as opposed to semi-direct or direct effects) and to help identify aerosol conditions from the lidar with greater confidence (i.e., higher signal/noise). Because we still have higher confidence in classifying aerosol conditions for the MOONLiT subset than in this new group, the old data are still presented, although now mostly in the supplementary material. Clouds with very different characteristics and less certain aerosol conditions (i.e., daytime clouds, optically thick clouds, icy clouds) were not included in this analysis. For more discussion on our reasoning, please see response #3iii below.

ii.  We have made a concerted effort to better clarify our goals and methods, which we think will also help address some of the reviewer's concerns (please see response #2 below).

iii. We have stressed more clearly that our conclusions are limited, and are only fully substantiated for the subset to which they pertain.

2) The methods have been clarified as follows:

i.   "The authors are left with only a handful of cases (It's not clear to me exactly how many, maybe thousands) from two years worth of satellite data."

The confusion on sample number might have arisen due to a typo in the formatting of Table 2 where the sample numbers were presented. The sample numbers (labeled in the "n" columns) for the "all cloud" cases were truncated, so that a sample number of clouds over sea ice that was actually 4579, for example, appeared as 45 and 79 separately. We have corrected this error in the new draft.

ii.  "It is very difficult to believe that the handful of cases examined here can be representative of meteorological conditions over the entire arctic and throughout the year."

We agree with this comment, and addressed it in several ways. First, we have changed the title to more clearly emphasize that our samples cover only nighttime data. This was previously discussed in the manuscript (e.g., p3, lines 1, 14, and 16; p 4 line 12; p 17 line 31 in the ACPD paper), but the change to the title will hopefully make it clearer to the reader that our data are relevant to mainly wintertime (and some spring and fall) cases, and not to conditions throughout the year.

Next, we now better clarify in the title, abstract, and in numerous places throughout text that the meteorological regimes discussed were only for regions over the Arctic Ocean and not for the entire Arctic. For example, please see p. 2, l. 20 & 24; p. 3, l. 24; p. 4, l. 26, etc.

Thirdly, as previously stated in the ACPD paper (e.g., p. 17, l. 27-35), our original conclusions were intended to only represent a subset of optically thin, predominantly liquid clouds, and not all nighttime Arctic Ocean clouds. We now more strongly emphasize this point throughout the text (e.g., first paragraph of the new section 3.2).

Lastly, regarding the general representativeness of our data, we have conducted a series of additional analyses to place our results in context of how much of the total Arctic region they cover (p. 10, l. 5-13; p. 16, l. 5- 12). The results of these analyses have helped narrow our maximum regional estimates of radiative impact to more precise levels, and are also used to more clearly stress the limitations of our conclusions in the abstract and elsewhere (e.g., p. 19, l. 28; p. 10, l. 5-13).

iii. "An example of this over-filtering of the data is eliminating clouds that are detected by CALIPSO but undetected by CloudSat;" and "By limiting analysis to cases where both CALIPSO and CloudSat identify approximately the same cloud height the authors throw out a great number of cases where clouds may have radar reflectivites below the detection threshold of the radar but are the thin liquid clouds of interest to the study.... I can't reconcile this with the statement that 95% of the data are included in the analysis."

Clouds that were detected by CALIPSO and not detected by CloudSat actually were included in the analysis. As can be seen in Table 2, the total data sample number is higher for parameters obtained from CALIPSO (e.g., cloud base height) than for parameters derived from CloudSat (e.g., cloud droplet effective radius), since there were no trustworthy CloudSat data in clouds that CloudSat did not detect.

The confusion here likely arose from hangover information from a previous draft that should have been deleted. The third row in the Table 1 CloudSat criteria has now been deleted. To improve clarity, we also added a footnote to Table 1 for the lines with CloudSat data, as follows:

**"*as available for clouds with radar reflectivity above the detection limit of -29 dBZ."**

Our wording in section 2.1 has also been edited for clarity on this issue (new text in bold).

"~~To ensure comparability of clouds measured with both instruments, only clouds for which the reported cloud top height was within 0.4 km in both instruments were included (i.e., ~95% of the data).~~ Because the CloudSat radar does not accurately estimate cloud properties below ~0.7-1 km agl (Huang et al., 2012; Mioche et al., 2015), we focused on clouds with bases ≥ 1 km agl. We recognize that many Arctic clouds lie below this altitude (Devasthale et al., 2011a; Shupe et al., 2011) and that these low-level clouds have important radiative impacts. However, we still chose to focus on clouds at these higher levels to obtain higher certainty in the data. **Also,**

**some of the very thin clouds detected by CALIPSO had radar reflectivities that were too low to be detected by CloudSat, and CloudSat may sometimes mistakenly assign precipitating ice as a cloud (de Boer et al., 2008). Therefore, radar reflectivity data and CloudSat reflectivity-derived cloud parameters, where available, were obtained from the height bins closest to where CALIPSO detected a cloud."**

The text stating that ~95% of the clouds had cloud top heights within 0.4 km of each other was a related error. The 95% number was calculated from the final dataset made after figuring out that it was best to approximate CloudSat cloud height bins from the CALIPSO cloud height bins; presenting this calculation was inconsistent with the proceeding sentence. So what the 95% number signifies is that in the cases where both CALIPSO and CloudSat observed a cloud in approximately the same height range, ~95% of the time, the observed cloud top height bin from CloudSat was within 0.4 km of the cloud top height bin that would have been expected based on the CALIPSO data. To avoid confusion, we removed this from the new text. Thanks for drawing our attention to this.

3) The reviewer voiced concern over the sample selection criteria. To paraphrase, they wanted to know why very rigorous cloud selection criteria are useful and necessary (i), especially given: (ii) that stricter criteria comes at the tradeoff of sample representativeness, (iii) that we might obtain other useful information by loosening the criteria to include various products or data, and (iv) that filtering may introduce sampling biases.

As previously mentioned, at the reviewer's suggestion, we have expanded the dataset to be more comprehensive and so hopefully some of these concerns are already addressed. However, even in the expanded dataset we still had to make choices on which clouds to include in the analysis, and so in this context we will respond to individual points raised by the reviewer.

i) Regarding why the rigorous sample selection criteria are useful:

The quantifiably correct identification of a subset of clean background clouds is a crucial step in our method, which we now more clearly state in the paper (p. 2, l. 26-31). Any expansion in the scope of the study comes at the cost of higher and less quantifiable error in our results. For example, we now include clouds below another cloud layer, but uncertainty in the aerosol classification for these cases from the lidar is much higher. Thus, air mass classification for those clouds is now more reliant on the model, making errors less quantifiable, which we now discuss. As another example, our data in section 3.1 suggest that at night, CALIPSO without FLEXPART misses ~33% of dilute aerosol layers. If we had included daytime samples where 2B-FLXHR-LIDAR estimates are more reliable, CALIPSO would have missed ~60% of dilute aerosol layers based on preliminary analysis. Ultimately, the criteria we chose were selected to balance representativeness with data quality, erring on the side of data quality.

ii) As the reviewer mentioned, the tradeoff of improved certainty in aerosol conditions is reduced sample representativeness. The reviewer questioned the utility of the study based on the low sample representativeness.

> This is a good point that we have worked hard to address. Besides including more samples and more quantitatively discussing representativeness of the samples in the new section 3.2, we now more clearly state both the limitations and the utility of our results (as previously mentioned, p. 19, l. 28; p. 10, l. 5-13). Also, all trends within the cloud subsets are discussed in the context of both statistical significance (which incorporates sample number as a factor) and meaningfulness (as discussed on a case-by-case basis in the text).

iii) The reviewer suggested expanding our dataset to include a) clouds with higher optical depth, b) cloud radiative forcing from the CloudSat 2B-FLXHR-LIDAR product, and c) clouds below 1 km.

> One of the goals of this work is to provide a foundation for expanding the study to other cloud types that were too complex to include in one manuscript. Although we have made an effort to include a larger and more representative cloud dataset in our revised analysis, understanding the aerosol indirect impacts on different types of clouds with lower confidence in the assignment of clean background conditions is quite complicated. In many cases, we felt it would best be done in a separate paper that can be fully dedicated to the substantial amount of additional caveats, analysis, and discussion required for exploring those data. Please see responses to specific reviewer comments below.

> a) Specifically regarding the following reviewer question: "…what sense does it make to require the cloud optical depth to be less than 3? …[D]on't we want to know if there are changes in the relative frequency of occurrence of these optically thick clouds in the presence of aerosol?"

> COD is required to be < 3 to enable the lidar to detect aerosol layers below the cloud.

> We agree that it would be interesting to test this in our dataset, for example by seeing if there is a difference in the relative frequency of optically thick clouds in cases where high aerosol was seen above the clouds and the model indicated high aerosol should have been present below the cloud. We plan to explore this topic as part of continuing work, but it is beyond the scope of the current paper. These other cloud types are not otherwise discussed in the paper, and the different analytical approach required and the much less-certain results that would be provided would result in a longer discussion with results not central to the rest of the paper.

> For the reviewer's interest, as mentioned in the paper, daytime MODIS COD observations from Coopman et al., 2016 in liquid phase Arctic clouds suggest that a frequency difference similar to what the reviewer mentions should be observable.

> b) The reviewer also wondered why we didn't use the CloudSat 2B-FLXHR-LIDAR

product.

This is a good question. The 2B-FLXHR-LIDAR version R04 product is a general product useful for the many conditions and cloud types across the world. However, it also has some limitations for the specific types of clouds that are discussed in this study, particularly for its COD and $r_{el}$ input information.  It was important to us to obtain the most accurate estimate of $r_{el}$ available because, based on Twomey effect expectations, we would expect that $r_{el}$ is one of the most important parameters affected by aerosol indirect impacts. When clouds are detected by CloudSat, the 2B-FLXHR-LIDAR product assigns cloud $r_e$ and COD from the CloudSat 2B-CWC-RO product. LWC is then estimated from these parameters for input into the model.  As discussed in section 2.1.2, the CloudSat 2B-CWC-RO product is associated with a lot of errors, particularly for $r_{el}$.  We reduced some of those errors in our study by only focusing on predominantly liquid clouds, and specifically selecting the 2B-CWC-RO subproduct that assumes droplets were liquid.  That was not done to our knowledge in the 2B-FLXHR-LIDAR version R04 product.

In clouds where no CloudSat data were available, the 2B-FLXHR-LIDAR product estimates $r_e$ based on temperature; above -20 $^{o}$C clouds are assumed to be liquid and assigned a $r_e$ of 13 μm.  Below that level clouds are assumed to be ice and assigned a $r_e$ of 30 μm (Henderson et al., 2013). Since all of our clouds were predominantly liquid-containing, but some of them reached temperatures well below -20 $^{o}$C (Table 2), the 2B-FLXHR-LIDAR $r_e$ temperature-based estimates are likely to contain higher errors than in our method for our specific cloud subset of interest. Unlike in the 2B-FLXHR-LIDAR product, we did not attempt to estimate $r_e$ in clouds where no CloudSat data were available.  Instead, our radiative flux calculations were based on the average for clouds with slightly thicker CODs where CloudSat data were detected (please see Table 2).

Another difference is that the CloudSat 2B-CWC-RO product determines COD when those clouds are detected by CloudSat, and otherwise it uses the CALIPSO COD values (at least for nighttime data when MODIS data are unavailable).  Because our particular sample set contained a relatively high fraction of cases with no CloudSat data, we thought it best to use CALIPSO CODs across all clouds to obtain a more internally comparable dataset.

Other than a few other minor differences, the radiative transfer calculations in the 2B-FLXHR-LIDAR calculations are not expected to be very different from our own. Like us, their model relies of surface conditions detected by passive microwave, and like us, they conducted two additional sets of flux calculations that are performed with all clouds and all aerosols removed, respectively. For these reasons, we chose to calculate radiative fluxes using our own method in this particular study.  We agree with the reviewer that the 2B-FLXHR-LIDAR product could be very useful in future studies that cover a wider group of Arctic cloud types (for example, daytime clouds), and in future work we hope to incorporate this product more.

c)   The reviewer suggested we include clouds in the lower 1 km of the atmosphere.

At the reviewer's suggestion we now include clouds with bases down to 200 m above the ocean surface in the new analysis, with some caveats. As mentioned in the paper, "the CloudSat radar does not accurately estimate cloud properties below ~0.7-1 km agl (Huang et al., 2012; Mioche et al., 2015)." Thus, the CloudSat data for shallow clouds only represent clouds > 750 m asl. We elected to keep the 1 km criterion in the MOONLiT portion of our study in order to base our reflectivity, $r_{el}$, and radiative transfer conclusions on a dataset in which we have higher confidence. We do not include clouds with bases below 200 m to avoid fog and to enable detection of below-cloud aerosol.

iv) The reviewer also expressed concern about the biases potentially induced by our sample selection criteria. We discussed these biases in detail in the former section 2.3 (now section 3.2). For the reasons discussed above, for the purposes of the present study, the cumulative errors induced by biases related to the sample selection criteria are likely to be much smaller than the error added by including cloud subsets with poor quality data or with uncertain aerosol influence. We now note this at the end of section 2.3.

4) Regarding clouds that are detected by CALIPSO but undetected by CloudSat, the reviewer asked: "Many of these clouds will be shallow liquid-only clouds with small drop sizes … Wouldn't one interesting test be to determine if these clouds are more prevalent in the polluted conditions? This might be expected from the author's hypothesis."

We thank the reviewer for the good suggestion. The reviewer is correct that there is a significantly higher probability of clean background clouds being detected by CloudSat than in all clouds and in aerosol-impacted clouds. We now include this information in the text.

SPECIFIC COMMENTS:

1) Section 2.1.2: By limiting analysis to cases where both CALIPSO and CloudSat identify approximately the same cloud height the authors throw out a great number of cases where clouds may have radar reflectivites below the detection threshold of the radar but are the thin liquid clouds of interest to the study. Eliminating clouds that have a base height greater than 1 km further aggravates this situation. In fact the authors have chosen a sampling strategy that minimizes the data availability from either instrument because it will be infrequent that clouds have optical depth less than 3 but still have a radar reflectivity above the -28 dBZ CloudSat sensitivity. This is why it looks likethere are maybe only a few hundred points on figure one. I can't reconcile this with the statement that 95% of the data are included in the analysis. How many pixels are included in the analysis? How many total pixels are there over the two year period?

This comment has already been addressed above. We actually did include clouds that CALIPSO observed but CloudSat did not (see general response #2iii). The sample

numbers for Figure 1 are noted in Table 2. Please see the response to general comment #3iv for more discussion on the inclusion of data below 1 km.

2) Page 5, line 31: Why exclude precipitation cases? Don't we expect some aerosol influence on the occurrence of precipitation?

As can be seen in Table 2, we noted the relative percent of precipitating clouds that otherwise met our sample criteria in each of the different air mass types, and there was an apparent aerosol influence. However, all other cloud characteristics listed in Table 2 are for clouds with no observed precipitation. To better explain this in the paper, we have added the following to the methods section:

"CloudSat may sometimes misclassify precipitating ice as part of the cloud (de Boer et al., 2008), which can lead to overestimation of $r_{el}$. Quality-flagged data were excluded, such as observations from precipitating clouds, as determined from the CloudSat 2B-CLDCLASS-LIDAR version R04 product. Note: although we counted the number of cases where precipitation occurred for comparison at a different step, precipitating cases were otherwise excluded **from most other derived cloud parameters in the analysis. These cases were excluded in order to obtain comparable data across cloud characteristics, which was particularly important for the longwave emissions calculations detailed in section 2.2 that included the $r_{el}$ as one of several input parameters.**"

And the following to footnote B in Table 2:

"Precipitating clouds were included in [the % precipitating] metric only; for all other attribute classifications, clouds were required to have no observed precipitation **in order to be comparable with $r_{el}$ estimates that were most reliable in non-precipitating clouds**."

3) Section 2.3: The authors seem to recognize that the artificial filters that they are applying to the data may well introduce biases. So why not include all the clouds regardless of optical depth or detection by radar?

As mentioned in general comment #2iii above, clouds not detected by radar were included in the original analysis. We also have supplied more information on our reasons for excluding clouds with high optical depth in our response to general comment #3iiia above. For our response on biases, please see the answer to general comment #2iii above.

4) Fig 3: Where does sea ice data come from?

As mentioned in the paper, "NOAA/NSIDC Climate Data Record of Passive Microwave Sea Ice Concentration, version 2 data (Meier et al., 2013; Peng et al., 2013) were used to approximate the fractional sea ice cover over ocean at the specific month and location

of each profile. A sample was classified as being primarily over sea ice or open ocean when the sea ice fraction at the given location and month was > 80% or < 20%, respectively."

5) Fig 3. Does this map include only the filtered data points shown in Fig 1.

Yes, it was obtained from the sea ice concentration below each cloud point in the study during the month that cloud was sampled. This has now been clarified in the Figure caption.

6) Page 11, Line 11: How is precipitation determined? Which product?

As mentioned in the methods section: "Cloud phase and precipitation occurrence were acquired from 2B-CLDCLASS-LIDAR version R04 estimates (Wang, 2013)."

7) Page 11, Line 22: I see Fig 1 differently. To my eye there is a clear clustering of the data with substantially more aerosol cases north of Europe and relatively more clean cases north of Siberia and North America. This statement is not justified by the analysis.

We have removed this sentence. Note: regional clustering of aerosol-influenced cases is more apparent in the larger dataset (see the new Fig. 1).

8) Page 12, Line 8: It is fairly obvious that you won't find an optical thickness difference when you have artificially limited the range of optical thicknesses to less than 3.

This sentence is no longer relevant in the new analysis.

---

## Author Comment (AC2) · 13 Apr 2017

Zamora et al. ACPD Reviewer Responses

Response to reviewer #2

GENERAL COMMENTS:

This manuscript by Zamora et al. presents an extensive study of thin liquid clouds over the Arctic and how these are affected by aerosol loading. The study combines satellite data from CALIPSO and CloudSat with FLEXPART modeling and aircraft measurements to better distinguish to which degree that the clouds were affected by aerosols. The study is limited to nighttime thin clouds between 1 and 8 km height and an estimation of the radiative impact of these clouds is provided. The manuscript is well written and contains detailed discussions regarding the uncertainties in the method and results. I recommend that the manuscript be published after answers to the following comments have been provided.

We thank the reviewer for their helpful comments, which have improved the paper.

SPECIFIC COMMENTS:

1) The study only includes nighttime clouds that have a COD < 3 and that are liquid. For the clouds to be included in the study they also must have an altitude between 1 and 8 km. In the methods section there are detailed descriptions of removal of data due to several other criteria considering confidence in data etc. My question is how representative the clouds included in the study are for the general conditions in the Arctic. Could you provide an estimate of how common these liquid clouds are? If the clouds in this study represents the conditions during 80% of the time or 20% of the time makes a big difference. I believe that the second sentence in the abstract may be a bit bold if it turns out that these clouds are not very common in the Arctic.

This was actually a very helpful suggestion, and in combination with comments from the first reviewer, it has helped us reframe the discussion and estimate radiative impacts in a more useful way. As requested, we now provide more information on how common these kinds of clouds are in context of the general conditions in the Arctic. To estimate cloud coverage, we examined the relative fraction of profiles containing our cloud subset vs. any cloud, and vs. the total profile number over the Arctic. The following addition to the text was added to the new section 3.2 (formerly section 2.3):

"It is important to emphasize that the ONLi cloud group is not representative of all Arctic clouds. During our study period, ONLi clouds were present in only 5.28% of all total comparable nighttime cloudy profiles over the Arctic Ocean ("comparable clouds" defined as having a satisfactory in-cloud CAD score of 70-100 and with cloud bases > 200 m to exclude fog). Liquid-dominated clouds tend to be found at lower altitudes than thicker opaque clouds and thus may not always be identified in multi-layer clouds using CALIPSO. However, even though the actual prevalence of these clouds may be somewhat underestimated, it is clear that ONLi clouds represent just a small fraction of all Arctic clouds. Thus, we emphasize that the aerosol indirect responses described in

this paper are not necessarily representative of general Arctic clouds."

The abstract has been re-written with this information being highlighted, and limitations on the analysis are now more fully discussed throughout the text (e.g., p. 20, l.1; p. 10, l. 10-18).

Another large change we made based on this comment was in how we estimated the radiative impacts to the surface. Before we had quantitative information on how common the clouds were, the maximum regional cloud longwave impacts were estimated by multiplying those expected in a 100% homogeneous cloud environment by the total cloud fraction of all clouds from a different study (Kay and L'Ecuyer, 2013). However, that was a very large over-estimate of the actual impacts since obviously (to us, in retrospect) our cloud subset is only a small portion of all clouds. Now with much more accurate information on the actual coverage of these specific clouds, we have provided a much reduced, and more precise and useful maximum regional impact. Thanks for getting us thinking about this! For more information on how these changes were implemented in the paper, please see section 3.6 (especially the first 2 paragraphs).

The reviewer also said, "The description of the data selections is very well written and detailed. However, it would be nice to know approximately how much data are lost at each step in the selection process."

At the suggestion of reviewer #1, we expanded the dataset to include many more types of optically thin, liquid clouds so that many of the previous steps in the selection process are no longer relevant (please see the new Table 1, for these changes). However, we still do refer to the previous MOONLiT cloud subset for reference, because this subset still has the highest certainty in aerosol classifications. Thus, for the reviewer's reference, we have added the following information in the Supplementary material:

"If the MOONLiT criteria were changed to include a) clouds with bases that were 200 m instead of 1 km above the surface, b) clouds above a separate ice cloud, or c) clouds below other non-opaque cloud layers (icy or otherwise), the MOONLiT cloud sample size would respectively have increased by 107 (121), 16 (28), and 303 (617)% over sea ice (open ocean). Any other differences between the MOONLiT cloud subset and the ONLi cloud subset was due to cases where uncertain aerosol CAD scores (<70) existed above or beneath cloud layer of interest. These clouds were allowed in the ONLi cloud subset, but not in the MOONLiT cloud subset."

Page 4, line 11: There are large land areas in parts of the described regions. Were these removed from the dataset?

Yes. That we focus only on clouds over the Arctic Ocean has now been clarified in the title, abstract, and throughout the text.

Page 4, line 22: Were all the cases averaged to 80km resolution or do the different cases have different resolutions?

Yes, for a cloud to have been present in a clean background air mass, the CALIPSO transect in which that cloud had been found had to have been horizontally averaged across 80-km with no evidence of an aerosol layer. The CALIPSO aerosol layer

algorithm works by first looking for evidence of an aerosol layer at 5-km resolution (where the strongest aerosol layers would be observed). If there is a weak aerosol signal, it might not be identifiable from the noise present at a 5-km resolution. Thus, if evidence of an aerosol signal is not found at 5-km resolution, the algorithm progressively lowers background noise by averaging over a larger area until a maximum of 80-km. Please see the first paragraph of section 2.1.1 for further information. To clarify this better in the text, we have changed the referred-to sentence as follows:

"The "clean, background" cloud subset met the above criteria, but no aerosol features were permitted above or below cloud **even when air masses had been** horizontal**ly** averaged **across** 80-km **in the CALIPSO aerosol detection algorithm, which is the resolution that detects weak aerosol layers with highest confidence**."

==Page 7, line 9: Why is data 10 degrees further south than the satellite data included in the comparison?==

To better answer this, we have changed the line in question as follows:

"**The aircraft data with highest aerosol particle concentrations were clustered between 50-60° N during this campaign. Thus, we included** aircraft data **from** between 50-82° N (subarctic + Arctic) **in order to assess comparable ranges of dilute and concentrated aerosols expected to be present over the greater Arctic**."

==Page 15, line 12: In the calculations of the indirect radiative effect of aerosols on MOONLiT clouds you write that you use the clean background cloud subset. Previously in the method you write that the parameters used in the calculations are cloud base height, cloud thickness and COD. For the cases over sea ice the COD is the same for the clean background and all cases datasets which means that the differences in the radiative effects is due to the difference in cloud base height (1.8 km vs. 1.9 km) and the difference in the cloud thickness (0.9 km vs. 1.2 km). Did I understand this correctly? Could you comment on this?==

This is mostly correct, except that observed cloud droplet effective radius was also used as a variable input parameter from the cloud dataset for the radiative impact calculations. To make it clearer to the reader which parameters were used in the calculations, the below information has now been re-arranged as follows:

"**Variable input parameters for the radiative impact calculations included** cloud base height, cloud thickness, COD, **and** $r_{el}$ **for** clouds over sea ice and open ocean. **Parameter values were taken from Table 2 median values, except for** $r_{el}$**, where the** interquartile range was used to reflect the larger uncertainty in that parameter."

For the reviewer's reference, in this instance, holding all other variables equal, aerosol-related changes in cloud optical depth were an order of magnitude more important for radiative effects than the changes in cloud droplet effective radius, and the changes in geometric thickness had nearly no impact, as now discussed in the text (p. 17, l.11-14).

==Figure text figure 3: "where a value of 0 indicates that the ocean surface was the next lowest feature". Does ocean surface here also mean sea ice?==

Yes, this is what we meant, as average Arctic sea ice is generally less than 3 m thick (e.g.

Zhang and Rothrock (2003)). To clarify, the text has been revised as follows:

"Figure 3: The data shown in a) and b) are weighted-average gridded maps of features below individual cloud points from Figure 1a for a) sea ice fraction, and b) height of the next lowest feature associated with individual cloud profiles, where a value of 0 indicates that the ocean surface **or sea ice** was the next lowest feature. Over open ocean, multi-layer clouds were much more common than over sea ice. Shown in c) is a boxplot indicating the cloud base heights (km) for single layer clouds over sea ice (grey) and open ocean (blue)."

TECHNICAL CORRECTIONS:

Page 12, line 7: optical thickness should be changed to COD.

Edited as recommended.

**References**

Amante, C. and Eakins, B. W.: ETOPO1 1 Arc-Minute Global Relief Model: Procedures, Data Sources and Analysis. NOAA Technical Memorandum NESDIS NGDC-24. National Geophysical Data Center, NOAA., , doi:10.7289/V5C8276M, 2009.

Zhang, J. and Rothrock, D. A.: Modeling Global Sea Ice with a Thickness and Enthalpy Distribution Model in Generalized Curvilinear Coordinates, Mon. Weather Rev., 131(5), 845–861, doi:10.1175/1520-0493(2003)131<0845:MGSIWA>2.0.CO;2, 2003.

---

## Author Comment (AC3) · 13 Apr 2017

Response to J.G. Guo

This manuscript is very interesting and constitutes a unique contribution to the better understanding of aerosol indirect effect on Arctic clouds, given the key findings revealed by the combined CALIOP-CLOUDSAT data. I have one minor comment the authors can consider in the revision, which is as follows:

Page 2, lines 6-7: "3) the complexity of cloud responses to aerosol type and amount,": At least the following two papers can be cited to benefit the readers.

Chen T.M., Guo J.P., Z. Li, C. Zhao, H. Liu, M. Cribb, F. Wang, and J. He. A CloudSat perspective on the cloud climatology and its association with aerosol perturbation in the vertical over East China, J. Atmos. Sci., 73, 3599–3616, doi:10.1175/JAS-D-15- 0309.1.2016.

Fan, J., Wang, Y., Rosenfeld, D., Liu, X.. Review of Aerosol–Cloud Interactions: Mech- anisms, Significance, and Challenges. Journal of the Atmospheric Sciences 73, 11, 4221-4252,2016.

We thank Dr. Guo for the interesting and relevant papers, and for their interest in this work. We have added the Fan et al. (2016) review as a reference.

---

## Author Response (AR2)

Dear Dr. Perring, Co-Editor,

Thank you for accepting our manuscript, pending minor edits, for publication! We have addressed your comments below:

1. In your references to Table 2, I don't see where you discuss what it means for a significant result to become insignificant with altitude binning. Perhaps I simply missed it…. And what do you interpret this loss of significance with binning to mean in a larger sense?

*Thanks for the suggestion. We have clarified what it means when a significant trend becomes insignificant by adding the following text to section 3.4 (new text in bold):*

"Here, **non-shallow clouds > 1.1 km were associated with a systematic decrease in** the cloud droplet effective radius as expected aerosol influence rose, and the estimated mode $r_{el}$ was respectively 10.3, 10.1, and 9.8 μm for the ONLi clean cloud, all cloud, and the aerosol-influenced cloud subsets…. …**Also note that decreases in $r_{el}$ were not significant in shallow clouds (Table 2). We hypothesize that shallow ONLi clouds may be subject to different meteorological forcing than non-shallow clouds >1.1 km, as discussed in section 3.3, and that this forcing might overwhelm cloud sensitivity to aerosols.**"

*And:*

"This possibility is supported by a small but significant increase in the portion of detected liquid phase clouds within sea ice clouds above 1.1 km (Tables 2 and S1). The trend in phase was not significant in MOONLiT cases (Table S3), **and as with $r_{el}$, it was also not significant in shallow clouds (Table 2). However,** only a strong trend in MOONiT cases would be significant due to a very small sample size**, and we have reason to suspect that differential meteorological forcing on shallow clouds might overwhelm cloud sensitivity to aerosols at lower altitudes**."

*And in section 3.5:*

"To **better understand any meteorological bias induced by** aerosol height differences between clean vs. non-clean clouds, but still retain a sample size from our 2-year dataset that is as informative as possible, we separated clouds found over open ocean into three cloud-base-height bins (Table S2), and summarized the resulting information in Table 2…. …**Altitude-related biases from aerosol vertical distributions can be one cause for the loss significant trends over the open ocean with altitude binning, indicated by the blue coloring in Table 2. A loss of significance might also be caused by differences in cloud-aerosol response with altitude, as is observed in shallow and non-shallow clouds over sea ice (section 3.4);**

**the general reduction in sampling when the data are stratified could also be a contributing factor**."

I also don't fully understand what altitude binning means in the context of this table. Do you mean that you divided the set of observations into various altitude bins and checked for a relationship in each bin separately?

*Yes. This has been clarified in the Table 2 text with a new footer (d) as follows:*

*"Blue indicates that when binned by altitude[d], significance was lost[e]."*

**[d]Samples were divided into altitude bins (< 1.1 and > 1.1 km over sea ice, and < 1.1, 1.1-3.2, and > 3.2 km over open ocean); significance was then assessed within each altitude bin, as with the non-binned data.**

2. p2, line 30: is in turn key --> is, in turn, key
   *Fixed as recommended.*

3. p3, line 24: It was vital to our method that --> it was vital that
   *Fixed as recommended.*

4. p3, line 30: we identified the subsets --> we identified subsets
   *Fixed as recommended.*

5. p4, lines 8-9: recommend rewording as "vertical resolution of 30 m within the layer (up to 8 km) where most predominantly liquid Arctic Ocean clouds were found."
   *Fixed as recommended.*

6. p4, line 11-12: I believe you mean to say that the horizontal resolution increases (or the pixel size decreases).
   *Yes, thank you. We changed "decreases" to "increases."*

7. p 5, line 12: I think a word is missing after "background molecular". Possibly you meant to say "background molecular scattering"?
   *Thanks for noticing that. We added the word "scattering".*

Apart from the responses to the Co-Editor, we also would like to mention some minor changes in the text that we would like to make. These changes do not have any impact on the results, and have only minor impacts on the conclusions, but seem worth including for clarity and completeness.

In the abstract and conclusions we clarified that the indirect effect we saw was a cooling effect, and that it did not include any cloud fraction effects, which were not assessed in this study.

Also, after the ACPD version of this paper went online, a relevant paper from Lohmann (2017) came out that drew our attention to fact that the "thermodynamic indirect effect" term has also recently been used in the literature to describe a different process than that

which we had been calling the thermodynamic indirect effect (our definition follows that which was used in Jackson et al. (2012)). To avoid confusion for future readers and to more accurately reflect the current state of knowledge in the field, we changed the "thermodynamic indirect effect" term in this manuscript to something more descriptive. Also, in the Lohmann (2017) paper we become aware of further evidence for the so-called "deactivation effect." We had mentioned this mechanism by description in the ACPD article as an alternative mechanism to we had called the "thermodynamic indirect effect" that could also explain some of our findings. However, we had not given this effect as much weight as we now think is merited, based on this new information. Because most of the changes below (in bold) involve re-arranging text to adjust the focus, rather than presentation of new information, we have noted substantially new information in red:

*Abstract:*

[revised manuscript text omitted]